# Understanding the Mechanisms of Fast Hyper-parameter Transfer

**Nikhil Ghosh**[1]    **Denny Wu**[1,2]    **Alberto Bietti**[1]
[1]Flatiron Institute    [2]New York University
{nghosh, dwu, abietti}@flatironinstitute.org

## Abstract

The growing scale of deep learning models has rendered exhaustive hyperparameter (HP) optimization prohibitively expensive. A promising solution is the use of scale-aware HPs, which can enable direct *transfer* of optimal settings from small-scale grid searches to large models with minimal performance loss. Such approaches are useful when the optimal settings converge "fast" enough with scale, so that transfer is asymptotically more efficient than direct tuning. While approaches like the Maximal Update Parameterization ($\mu$P) have empirically exhibited *fast transfer* when scaling model width, a deeper conceptual understanding of the mechanisms enabling this behavior is still missing. Our work develops a conceptual framework for analyzing this phenomenon across different synthetic and practical scenarios. In synthetic settings, we present various examples where transfer either offers a provable computational advantage or fails even under $\mu$P. We then propose a key property that enables fast transfer in practice: by decomposing the optimization trajectory, the final loss splits into (i) a width-stable component that determines the optimal HPs and (ii) a width-sensitive component that continues to reduce loss but minimally affects HP choice. We conjecture that this condition explains fast transfer and empirically validate it holds in large language model pretraining.

## 1 Introduction

**Scaling-aware hyperparameters.**    A central dogma in empirical deep learning is that performance will steadily improve [KMH+20; HBM+22] as training data and parameter count increase. This paradigm incentivizes practitioners to train increasingly larger models while first conducting experiments at smaller, more economical scales. For such small-scale experimentation to effectively inform larger training runs, it becomes crucial to reason about hyperparameters (HPs) in a scaling-aware framework and to analyze the behavior of the *sequence* of progressively scaled-up training runs. For example, when scaling the width $n$ of a neural network, we should conceptualize the learning rate as the product of a scale-independent HP $\eta$ and a scaling factor $n^{-a}$. This perspective was initially formalized in the Tensor Program series [YH21; YHB+22; YL23] for the width-scaling of neural networks. It was shown theoretically that the correct scaling for ensuring "optimal" training in the infinite-width limit is the Maximal Update Parameterization [YH21, $\mu$P]. This parameterization specifies per-layer initialization variance and learning rate scalings so that as $n \to \infty$, the network activations and their updates are $\Theta(1)$, ensuring that the optimal $\eta$ is asymptotically scale-independent.

**Hyperparameter transfer.**    Empirical evidence [YHB+22; Lin24] has demonstrated that in fact, $\mu$P enables a much stronger property which we refer to as *fast hyperparameter transfer* (see Section 3) that is not apparent from its theoretical derivation. At a high level, fast HP transfer occurs when optimal HPs converge significantly faster with respect to width than the performance measure of interest. This phenomenon allows practitioners to perform HP selection on smaller proxy models and subsequently apply these optimal values to larger-scale training runs, drastically reducing the cost of HP tuning. Despite its practical utility, this phenomenon remains primarily an empirical success, with limited theoretical understanding of its underlying mechanisms.

**Our contributions.**    We provide insight into the puzzle of fast HP transfer by developing a framework for reasoning about HP transfer in Section 2. Then in Section 3 we define fast HP transfer in terms of convergence rates of optimal HPs and the loss, and connect it to the "usefulness" of transfer when performing compute-optimal grid search. We quantify these convergence rates in synthetic settings, and demonstrate that while valid in certain linear model, fast transfer is not guaranteed in neural networks even when using $\mu$P. Such examples illustrate that the benefits of transfer heavily depend on structural properties of the training process emerging from data, optimizer, and architecture.

To illuminate this structure, in Section 4.1 we introduce a novel *loss decomposition* obtained by decomposing the step-wise linearized loss change along the training trajectory. We empirically demonstrate that the linearization is an effective proxy for the true loss change when considering the *exponential moving average* (EMA) of the optimization trajectory. The faithfulness of this approximation is due to the smoothness of the resulting EMA trajectory (see Figure 7). Using the linearization, we track the loss change arising from the projection of the update in the top-$k$ directions that maximize loss change at each matrix layer. This decomposes the total loss change into two components: the *top-$k$ loss* arising from the dominant $k$ components and the remaining *residual loss*.

We conjecture that the training behavior on the top-$k$ subspaces concentrates quickly with width while the remaining directions provide performance gains as the width increases. In Figure 1 we validate this intuition by applying the loss decomposition introduced in Section 4 to a sequence of Llama-style transformers of increasing width trained under $\mu$P. We observe that the leading top-$k$ loss remains approximately invariant across widths, whereas the residual loss consistently improves with width, especially later in training. Moreover, since the top-$k$ loss provides the majority of the loss decrease, it is reasonable to

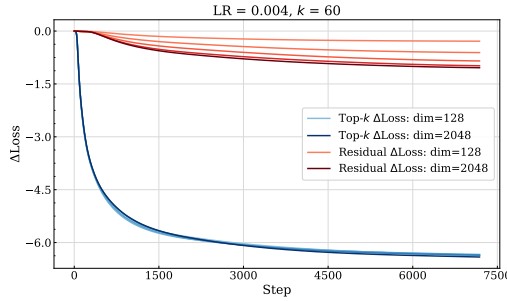

Figure 1: Loss decomposition into top-$k$ and residual components in transformers with varying widths.

expect that the optimal learning rate for the total loss will not deviate much from the optimal learning rate for the top-$k$ loss. Therefore, for an appropriately chosen $k < n$, the top-$k$ loss decomposition provides the following potential explanation for the phenomenon of fast transfer.

---

**Hypothesis 1: Fast Transfer via Loss Decomposition**

1. The top-$k$ loss converges rapidly with scale $n$, hence so do the optimal HPs for top-$k$ loss.

2. The optimal HPs for top-$k$ loss approximately *determine* the optimal HPs for the total loss.

---

In the rest of this paper, we provide a formalization for this intuitive explanation and use it to provide conceptual insights into the fast transfer phenomenon with support from experiments across synthetic and realistic settings, including large language model (LLM) training.

## 1.1 RELATED WORK

**Hyperparameter transfer.** The concept of hyperparameter transfer was introduced by [YHB+22], who showed that HPs tuned on small proxy models can transfer reliably to much larger ones under $\mu$P scaling. Subsequent empirical studies confirmed the effectiveness of learning rate transfer in transformers, while also highlighting certain limitations [Lin24; VBF25]. As [YHB+22] noted, the success of hyperparameter transfer is not explained from first principles. Addressing this gap empirically, [NMHO24] showed that top Hessian eigenvalues converge rapidly under $\mu$P across widths, suggesting a width-stable curvature that could underpin transfer. However, the connection between these spectral statistics and the optimal learning rate remain unclear. More recently, the concurrent work of [HW25] established a scale separation between certain macro- and micro-variables, thereby suggesting HPs can be effectively tuned in early training stages. In contrast to these results, our work explicitly defines (via a trajectory-level loss-decomposition) and connects the fast convergence of certain statistics of the optimization path to fast convergence of optimal HPs.

**Optimization trajectories.** Our fast transfer hypothesis rests on the existence of low-dimensional structure in optimization trajectories. [GARD18] observed that gradients rapidly align with top eigenvectors of the Hessian, suggesting that GD operates within a "tiny subspace." [SAY24] refined this view, showing that while gradient variance concentrates in the top eigenspace, motion in these directions primarily drive oscillations rather than loss reduction. This behavior is consistent with edge-of-stability (EoS) [CKL+21; DNL22] and motivates our use of EMA smoothing: by averaging out oscillatory components tied to the top eigenspace, the smoothed trajectory reveals the low-dimensional subspace where genuine loss decrease occurs. Similar low-dimensional structure also appeared in theoretical settings on gradient-based feature learning, such as the learning of multi-index models [AAM22; DLS22; BBPV23; MHPG+23; GWB25], where SGD "localizes" parameters into

low-dimensional subspaces. Recent works have also shown that gradient updates induce a spiked (signal-plus-noise) eigenstructure in the conjugate kernel [BES$^+$22; MLHD23; WWF24; DPC$^+$24] or the Hessian [BAGHJ23; MW25] of the neural network, and the top eigenvectors contain information of the target function. Drawing inspiration from these theoretical settings, we explicitly isolate the width-invariant structure in realistic settings using a spectral decomposition and link it to HP transfer.

## 2 PRELIMINARIES AND FORMAL FRAMEWORK

Let $n$ denote the scaling dimension. The scaled HPs used during training at scale $n$ are:
$$\mathcal{H}_n(\boldsymbol{\nu}, \boldsymbol{\gamma}) = (\nu_1 n^{-\gamma_1}, \ldots, \nu_h n^{-\gamma_h}),$$
where we refer to $\boldsymbol{\nu} = (\nu_1, \ldots, \nu_h)$ as the HPs and to $\boldsymbol{\gamma} = (\gamma_1, \ldots, \gamma_h)$ as the scaling exponents. Conceptually, $\boldsymbol{\nu}$ are a set of $n$-independent constants which are tuned for a specific problem and $\boldsymbol{\gamma}$ are a set of exponents which specify how to scale the HPs with $n$ so that training can be performed at any scale $n$ using $\mathcal{H}_n(\boldsymbol{\nu}, \boldsymbol{\gamma})$. In this paper, we focus on the "optimization hyperparameters" of the abcd-parameterizations of [YL23] (see Appendix B). This setting encompasses the width scaling of hyperparameters needed for training standard neural network architectures using common optimizers such as SGD and Adam. Thus we will often simply refer to the scale $n$ as the width; however our framework can be used more broadly for reasoning about scale-aware HPs.

In our setting the *training procedure* $\mathcal{A}$ is held fixed and we only vary $n$, the HPs $\boldsymbol{\nu}$, and the scaling $\boldsymbol{\gamma}$. The training procedure returns a stochastic output $\mathcal{A}(n, \boldsymbol{\nu}, \boldsymbol{\gamma})$. For neural network training this means that the architecture, optimizer, dataset, etc. are all fixed, and we will let $\mathcal{A}(n, \boldsymbol{\nu}, \boldsymbol{\gamma})$ be the resulting optimization trajectory. For a given configuration, we can measure a scalar metric $\phi_n(\boldsymbol{\nu}; \boldsymbol{\gamma})$ obtained from the output of $\mathcal{A}$ (e.g., the final validation loss). We tune HPs over a *search space* $\mathcal{X}$ which we take to be a $h$-dimensional box. We define the *optimal HPs* and the corresponding *optimal value* as
$$\boldsymbol{\nu}^\star(n; \boldsymbol{\gamma}) = \arg\min_{\boldsymbol{\nu} \in \mathcal{X}} \phi_n(\boldsymbol{\nu}; \boldsymbol{\gamma}), \quad \phi_n^\star(\boldsymbol{\gamma}) := \phi_n(\boldsymbol{\nu}^\star(n; \boldsymbol{\gamma}); \boldsymbol{\gamma}).$$
We omit the exponents $\boldsymbol{\gamma}$ when context is clear, and abbreviate the above as $\phi_n(\cdot)$, $\boldsymbol{\nu}^\star(n)$, and $\phi_n^\star$.

**Weak Transfer.** A basic requirement for a parameterization $\boldsymbol{\gamma}$ is that both the optimal HPs $\boldsymbol{\nu}^\star(n)$ and the local sensitivity of $\phi_n$ around $\boldsymbol{\nu}^\star(n)$ are (asymptotically) independent of the scale $n$. This guarantees that grid search over a fixed, scale-independent grid of HPs achieves near-optimal performance for all $n$. To ensure that relevant asymptotic quantities converge to well-defined limits, we impose regularity assumptions on $\{\phi_n\}$ so that a well-defined limit $\phi_\infty$ exists and the optimal HPs $\boldsymbol{\nu}^\star(n)$ converge to the minimizer $\boldsymbol{\nu}^\star(\infty)$ of $\phi_\infty$. To quantify the transfer suboptimality, we further assume that $\phi_\infty$ is **locally strongly convex**. This condition links the suboptimality of a transferred HP to loss suboptimality and ensures that performance meaningfully degrades away from the optimum. Without such a condition, $\phi_\infty$ can be flat near its minimizer, making accurate HP selection irrelevant.

We will say that the parameterization $\boldsymbol{\gamma}$ admits *weak transfer* (or simply that "HP transfer holds") when the above conditions are satisfied. The rigorous formulation is given in Definition 3 (Appendix B). We regard this as the "weakest" notion of HP transfer, since it concerns only the asymptotic convergence of quantities rather than their convergence rates. Prior work [YHB$^+$22; YYZH23] has argued informally that only "optimal" parameterizations can admit weak HP transfer and that $\mu$P is the unique optimal parameterization for general tasks. We formalize the first claim in Theorem 9 and assume that weak transfer holds under $\mu$P.

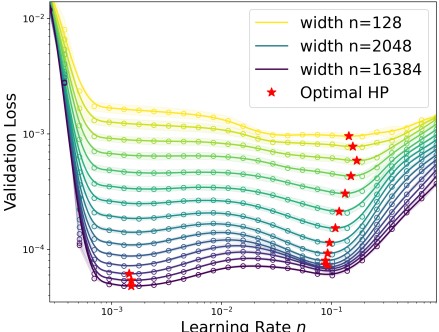

Figure 2: Learning Gaussian $k$-index model using two-layer ReLU network with $\mu$P. The HP of interest is the Adam learning rate – see Appendix E.1 for details. The optimal HP exhibits an abrupt shift at $n = 8192$.

Note that weak transfer alone does not imply that transfer under $\mu$P is actually *useful*, as it does not quantify the convergence rates and computational gain over direct tuning. In particular, even though the optimal HP converges to a well-defined limit, if such convergence happens very slowly, then one cannot reliably infer $\boldsymbol{\nu}^\star(\infty)$ from small-scale proxies — see Figure 2 for a numerical example, where the optimal HP up to width $n = 8192$ becomes clearly suboptimal for larger-width models. In the next section, we will show that stronger conditions on the convergence rates are required for fast and useful HP transfer.

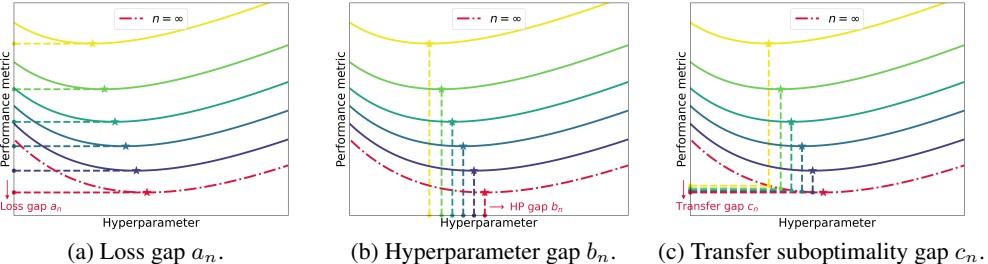

(a) Loss gap $a_n$.  (b) Hyperparameter gap $b_n$.  (c) Transfer suboptimality gap $c_n$.

Figure 3: Illustration of convergent quantities in Definition 1; note that $c_n$ captures the efficiency of HP transfer.

## 3  THE PUZZLE OF FAST & USEFUL TRANSFER

In this section, we relate the convergence rate of optimal HPs (e.g., learning rate) to that of the evaluation metric (e.g., validation loss). As previously discussed, the definition of weak transfer ensures that $\nu^\star(n) \to \nu^\star(\infty)$, but we need to characterize the rate of convergence in order to conclude the computational efficiency of the transfer strategy. As we will see in Proposition 1, the convergence rate of $\phi_n \to \phi_\infty$ already implies an upper bound on the convergence rate of the minimizers $\nu^\star(n)$ due to the local strong convexity condition of $\phi_\infty$. Moreover, it turns out that this convergence also informs whether it is computationally more efficient to use HP transfer than to directly tune the model at a single width $n$, as we show in Theorem 2. We then present toy examples where the convergence rates of the optimal HP and loss can be quantified.

### 3.1  CONVERGENCE RATES OF HP TRANSFER

**Notation.**  Given deterministic sequences $x_n \in \mathbb{R}$ and $y_n \in \mathbb{R}^+$, we write $x_n = O(y_n)$ to mean there exists $c > 0$ such that $|x_n| \le cy_n$ for all large $n$; $o(\cdot), \Theta(\cdot)$ are defined analogously. We write $x_n \sim y_n$ to mean $x_n = \Theta(y_n)$. We overload the same notation for random quantities: whenever $x_n$ is random, $O(\cdot), o(\cdot), \Theta(\cdot)$, and $\sim$ are understood in the corresponding stochastic sense (i.e., $O_p, o_p, \Theta_p$), and limits $x_n \to x$ are interpreted as convergence in probability unless stated otherwise.

**Tracking the Width Dependence.**  We now introduce three quantities that track the convergence of the optimal loss and HPs, illustrated in Figure 3.

**Definition 1** (Convergent Quantities)**.**  *We define the following width-dependent quantities.*

- ***Loss gap:*** $a_n = |\phi_n^\star - \phi_\infty^\star|$. *The quantity measures the discrepancy between the optimal value of the metric $\phi$ at the finite-width model (over all HPs) and that of the infinite-width model.*

- ***Hyperparameter gap:*** $b_n = \|\nu^\star(n) - \nu^\star(\infty)\|$. *The quantity measures the discrepancy between the optimal HPs at width $n$ and the infinite-width counterpart.*

- ***Transfer suboptimality gap:*** $c_n = |\phi_\infty(\nu^\star(n)) - \phi_\infty^\star|$. *The quantity measures the performance gap between two infinite-width models: one with the optimal HP tuned at infinite-width, and the other with the optimal HP at a smaller width.*

**Fast Transfer.**  In the following, in addition to assuming HP transfer holds (Def. 3 in Appendix C.1) we will make the mild technical assumption that the "uniform loss gap is locally tight" which implies that $a_n \sim \|\phi_n - \phi_\infty\|_{\sup}$ (see Def. 4 and Lemma 12 in Appendix C.2). The following proposition uses strong convexity around the optimal infinite-width HP to directly connect the convergence rates of the optimal HP and the evaluation metric (see Appendix C.2).

**Proposition 1.**  $b_n = O\big(a_n^{1/2}\big)$ *and* $c_n = \Theta(b_n^2) = O(a_n)$.

Since HP transfer holds, we have by definition $a_n, b_n, c_n \to 0$ as $n \to \infty$. This being said, *a priori* we may have $b_n = \Theta\big(a_n^{1/2}\big)$ and hence $c_n = \Theta(a_n)$. This would indicate that the transfer suboptimality gap converges at the same rate as the loss gap. In practice however, we tend to observe a much faster rate for $c_n$ relative to $a_n$. We refer to this phenomenon formally as *fast transfer*.

**Definition 2.**  *We say that HP transfer is fast if* $c_n = o(a_n)$ *which occurs iff* $b_n = o\big(a_n^{1/2}\big)$.

For instance, if $a_n \sim n^{-\alpha}$ and $b_n \sim n^{-\beta}$, then fast transfer occurs if and only if $\beta > \alpha/2$.

**Useful Transfer.**  The notion of fast transfer also coincides with the computational usefulness of HP transfer when performing a grid search. Consider two HP tuning strategies: **direct** performs grid

search on models of a single width, and **transfer** performs grid search on models of a smaller width and then trains a final model of larger width using the obtained optimal HPs. Suppose we have a compute budget of $\mathcal{F}$ flops, and the amount of flops needed for a single training run scales with $n^r$ for models of width $n$, where $r = 2$ for standard optimization algorithms on standard architectures. We say that transfer is *useful* if the transfer strategy yields better performance than the direct strategy for a given compute budget. The following theorem characterizes when this occurs.

**Theorem 2** (Useful transfer). *Suppose $a_n \sim n^{-\alpha}$ and $b_n \sim n^{-\beta}$. We are given a compute budget of $\mathcal{F}$ flops, we tune $h$ different HPs, and a single training run at width $n$ costs $n^r$ flops.*

(a) ***Direct tuning.*** *If we directly conduct grid search on a width-$n$ model, the compute-optimal performance scales as $\mathcal{F}^{\frac{-2\alpha}{h\alpha+2r}}$ and is obtained at width $n^\star \sim \mathcal{F}^{\frac{2}{h\alpha+2r}}$.*

(b) ***Transfer.*** *If we grid search over width-$n$ models, then transfer to width-$M$ model, the compute-optimal loss scales as $\mathcal{F}^{\frac{-\alpha}{r}} + \mathcal{F}^{\frac{-2\beta}{h\beta+r}}$, obtained at widths $n^\star \sim \mathcal{F}^{\frac{1}{h\beta+r}}$ and $M^\star \sim \mathcal{F}^{1/r}$.*

*Hence transfer is useful if and only if $\beta > \alpha/2$, i.e., $b_n = o(\sqrt{a_n})$.*

Observe that the requirement $\beta > \alpha/2$ is the same condition as *fast transfer* (Definition 2). Hence under local strong convexity of $\phi_\infty$ with respect to $\nu$, the naive loss convergence rate already implies that transfer never underperforms the direct tuning strategy asymptotically, provided that the HPs are parameterized to be $n$-independent. While this supports the effectiveness of $\mu$-transfer [YHB+22], it does not address the question of when the optimal HPs converge faster than what is implied naively by the loss convergence. The question turns out to be subtle, and the following subsection presents simple examples where fast transfer may be present or absent.

## 3.2 Examples of Fast and Slow HP Transfer

The precise scaling of quantities in Definition 1 is infeasible to measure in large-scale scenarios: since the infinite-scale model ($n \to \infty$) is inaccessible, we must resort to power-law fits from finite-$n$ data, which requires a fine grid search of HPs at each scale. Therefore, in this section we present synthetic settings where the convergence rate of $(a_n, b_n, c_n)$ can be either analytically derived or reliably estimated from data. These settings allow us to quantify the computational gain of HP transfer.

**Fast HP Transfer: Random Features Regression.** First we consider tuning the ridge penalty $\lambda$ in a high-dimensional random features (RF) model with nonlinearity $\sigma$, where the target function is a single-index model with link function $\sigma_*$ on isotropic Gaussian input in $\mathbb{R}^d$. We aim to select the optimal regularization parameter $\lambda \in \mathbb{R}$ that minimizes the prediction risk $\mathcal{R}(f) = \mathbb{E}_{\boldsymbol{x}}[(y - f(\boldsymbol{x}))^2]$.

In the proportional limit where the number of training data $N$, dimension of input features $d$, and model width $n$ all diverge $N, d, n \to \infty, N/d \to \psi_1, n/d \to \psi_2$ where $\psi_1, \psi_2 \in (0, \infty)$, [MM22; GLK+20] derived precise asymptotics of prediction risk under standard assumptions (see Assumption 2). In this model, the number of trainable parameters is controlled by the ratio $\psi_2 = n/d$, and the infinite-width model is obtained by sending $\psi_2 \to \infty$. The following proposition quantifies the convergence of prediction risk and hyperparameter as a function of $\psi_2$.

**Theorem 3.** *Define $\lambda^*(\psi_2) := \arg\min_{\lambda \in \mathbb{R}} \mathcal{R}_{\psi_2}(\lambda)$, where $\mathcal{R}_{\psi_2}(\lambda)$ is the asymptotic prediction risk of the RF model with width $n/d = \psi_2$ and ridge penalty $\lambda$. Then under Assumption 2, we have*

- ***Loss gap:*** $|\mathcal{R}_{\psi_2}(\lambda^*(\psi_2)) - \mathcal{R}_\infty(\lambda^*(\infty))| = \Theta(\psi_2^{-1})$.

- ***Hyperparameter gap:*** $|\lambda^*(\psi_2) - \lambda^*(\infty)| = O(\psi_2^{-1})$.

- ***Suboptimality gap:*** $|\mathcal{R}_\infty(\lambda^*(\psi_2)) - \mathcal{R}_\infty(\lambda^*(\infty))| = O(\psi_2^{-2})$.

This theorem states that both the loss gap $a_n$ and the HP gap $b_n$ scale as $d/n = \psi_2^{-1}$, whereas the transfer suboptimality gap scales as $c_n \sim \psi_2^{-2} \ll a_n$. We conclude that the ridge regularization parameter $\lambda$ exhibits fast transfer per Definition 2. Consequently, Theorem 2 implies that tuning the ridge penalty on a small (narrower) RF model and then transfer is more compute-efficient than directly tuning the large-width model. To our knowledge, this gives the first concrete setting where the HP transfer strategy in [YYZH23] provably offers a computational advantage.

Figure 4a presents the prediction risk of the RF ridge estimator across varying widths $n/d = \psi_2$. We set $\sigma = \tanh, \sigma_* = \text{ReLU}, \psi_1 = 4, \sigma_\varepsilon = 1/4$. The analytical curves are obtained by solving the coupled Stieltjes transforms (9) and (10). Figure 4b confirms the scaling of $a_n, b_n, c_n$ in Theorem 3.

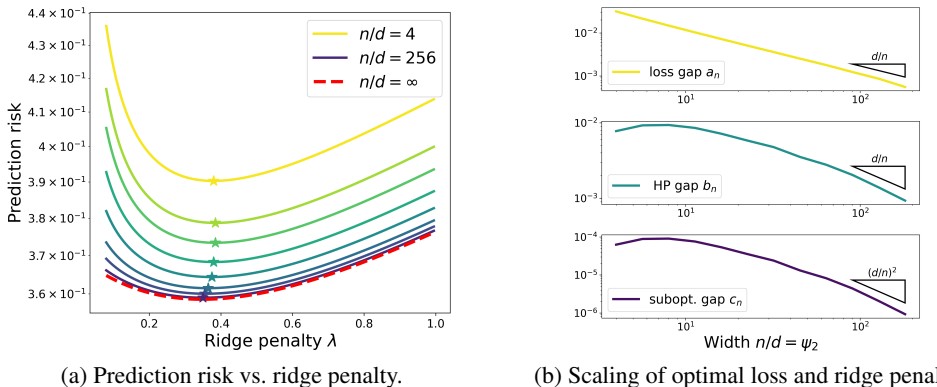

(a) Prediction risk vs. ridge penalty.      (b) Scaling of optimal loss and ridge penalty.

Figure 4: Optimal ridge penalty (generalization error) for RF regression to learn a single-index model.

**Slow HP Transfer: Two-layer ReLU Network.** Next we consider a classification setting with a shallow ReLU neural network: $f(\boldsymbol{x}) = \sum_{i=1}^{n} a_i \sigma(\langle \boldsymbol{w}_i, \boldsymbol{x} \rangle + b_i)$, where we aim to tune the learning rate $\eta$ that minimizes the validation loss. We set the target to be the norm indicator function, which is a well-studied function that requires a wide two-layer network to approximate [SL22],

$$ y = \mathbb{1}\big\{ \|\boldsymbol{x}\|_2^2 > F_{\chi_d^2}^{-1}(0.5) \big\}, \quad \text{where } \boldsymbol{x} \sim \mathcal{N}(0, \boldsymbol{I}_d), $$

where $F_{\chi_d^2}^{-1}(0.5)$ represents the median of a chi-square distribution with $d$ degrees of freedom. This threshold ensures that the classes are exactly balanced. We set $d = 2^6, n = 2^{14}$, and run the Adam optimizer [KB14] for $T = 2^{14}$ steps with batch size $2^8$ to minimize the binary cross-entropy loss. The initialization and learning rate are set according to $\mu$P [YH21].

In Figure 5a we observe that under $\mu$P, while the optimal learning rate admits a well-defined limit, there is still a visible drift towards the right. Moreover, Figure 5b illustrates that under a power-law fit, the optimal $\eta$ converges slower than the validation loss, and the estimated scaling $b_n \sim \sqrt{a_n}$ suggests that the HP convergence rate does not beat the agnostic rate from strong convexity. An interesting future direction is to analytically derive the scaling exponents in $a_n, b_n, c_n$ for $\mu$P examples and quantify the efficiency of learning rate transfer (beyond the definition of weak transfer in Section 2).

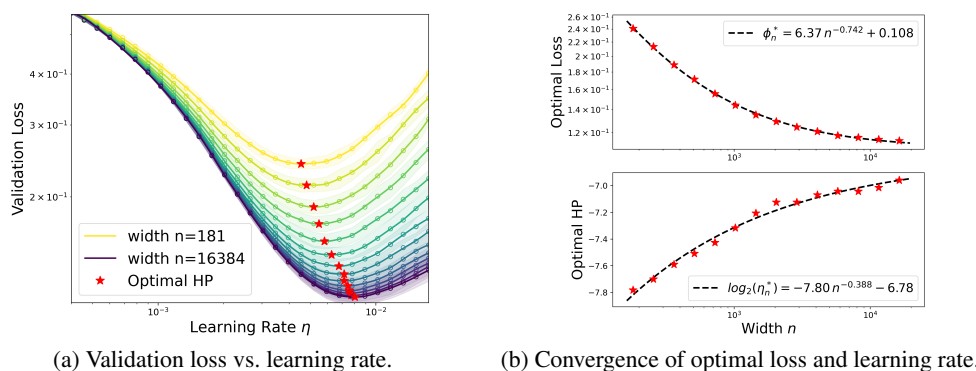

(a) Validation loss vs. learning rate.      (b) Convergence of optimal loss and learning rate.

Figure 5: Optimal learning rate (validation loss) for two-layer ReLU network to learn the ball indicator function.

## 4   FAST TRANSFER VIA TRAJECTORY DECOMPOSITION

Section 3 demonstrates that without further assumptions, the existence of a scale-independent limit of optimal HP (i.e., *weak transfer*) does not entail that transfer is more computationally efficient than direct tuning. At a high-level, for useful transfer to occur in practical neural network optimization settings, the optimal HPs should depend on some statistics of the training trajectory that converge faster than statistics tied to the performance metric of interest. In this section, we explicitly extract the fast-converging statistics from the trajectory and connect them to the optimal HPs. To do so, we will leverage prior intuition on the *low-dimensionality* of optimization trajectories (see Section 1.1): intuitively speaking, we may expect the movement in a small number of HP-sensitive directions to significantly contribute to the loss decrease and approximately determine the optimal HP. If the

associated low-dimensional statistics converge sufficiently fast, then the optimal HP also becomes stable across scale. For instance, it is plausible that the loss depends on all eigenvalues of the Hessian, whereas the learning rate is decided by the top components alone (which may converge faster with scale). In the ensuing subsection, we introduce a novel layer-wise spectral decomposition to identify the low-dimensional structure that informs the HP selection.

### 4.1 Top-$k$ Loss Decomposition

Let us consider an optimization trajectory $\boldsymbol{\omega} = (\boldsymbol{w}_0, \ldots, \boldsymbol{w}_T)$ of a neural network. Define the one-step loss change $\delta\mathcal{L}(\boldsymbol{w}_t) := \mathcal{L}(\boldsymbol{w}_{t+1}) - \mathcal{L}(\boldsymbol{w}_t)$, so that the overall loss change is the sum

$$\Delta\mathcal{L}(\boldsymbol{\omega}) := \mathcal{L}(\boldsymbol{w}_T) - \mathcal{L}(\boldsymbol{w}_0) = \sum_{t=0}^{T-1} \delta\mathcal{L}(\boldsymbol{w}_t).$$

Let $\boldsymbol{g}_t := \boldsymbol{\nabla}\mathcal{L}(\boldsymbol{w}_t)$ and $\delta\boldsymbol{w}_t := \boldsymbol{w}_{t+1} - \boldsymbol{w}_t$. Let us define $\delta\phi(\boldsymbol{w}_t) := \langle \boldsymbol{g}_t, \delta\boldsymbol{w}_t \rangle$ to be the linearization of $\delta\mathcal{L}(\boldsymbol{w}_t)$. Under appropriate smoothness conditions on the trajectory,

$$\delta\phi(\boldsymbol{w}_t) \approx \delta\mathcal{L}(\boldsymbol{w}_t) \quad \text{and} \quad \phi(\boldsymbol{\omega}) := \sum_{t=0}^{T-1} \delta\phi(\boldsymbol{w}_t) \approx \Delta\mathcal{L}(\boldsymbol{\omega}). \tag{1}$$

To ensure that such smoothness conditions hold and $\phi(\boldsymbol{\omega})$ is a useful proxy for $\Delta\mathcal{L}(\boldsymbol{\omega})$, we take $\boldsymbol{\omega}$ to be an *exponential moving average* (EMA) of the base optimization trajectory to remove oscillatory behavior. On realistic problems, we obtain excellent agreement between $\phi(\boldsymbol{\omega})$ and $\Delta\mathcal{L}(\boldsymbol{\omega})$ for EMA trajectories which perform at least as well as the corresponding base trajectories (see Fig. 7).

The utility of considering the linearization $\phi(\boldsymbol{\omega})$ in our setting is that we can decompose $\delta\phi(\boldsymbol{w}_t)$ based on the structure of $(\boldsymbol{g}_t, \delta\boldsymbol{w}_t)$ and recover a resulting decomposition of $\phi(\boldsymbol{\omega})$. Let us consider a fixed time index $t$. If $\boldsymbol{w} = (\boldsymbol{W}^{(1)}, \ldots, \boldsymbol{W}^{(L)})$ are the model parameters with corresponding gradients $\boldsymbol{g} = (\boldsymbol{G}^{(1)}, \ldots, \boldsymbol{G}^{(L)})$ such that $\boldsymbol{W}^{(\ell)}$ and $\boldsymbol{G}^{(\ell)}$ are tensors of the same shape, then

$$\langle \boldsymbol{g}, \delta\boldsymbol{w} \rangle = \sum_{\ell \in [L]} \left\langle \boldsymbol{G}^{(\ell)}, \delta\boldsymbol{W}^{(\ell)} \right\rangle.$$

Consider a single summand coming from $\boldsymbol{W} \in \mathbb{R}^{m \times n}$ with corresponding gradient $\boldsymbol{G} \in \mathbb{R}^{m \times n}$. To isolate the dominant directions of descent, we analyze the alignment between the gradient and the update via the *alignment matrix* which is the symmetrized product:

$$\mathcal{S}(\boldsymbol{G}, \delta\boldsymbol{W}) := \tfrac{1}{2}(\boldsymbol{G}^\top \cdot \delta\boldsymbol{W} + \delta\boldsymbol{W}^\top \cdot \boldsymbol{G}). \tag{2}$$

If $\mathcal{S}(\boldsymbol{G}, \delta\boldsymbol{W})$ has eigenvalues $\lambda_1, \ldots, \lambda_n$ such that $|\lambda_1| \geq \cdots \geq |\lambda_n|$, we define the top-$k$ stepwise linear loss change $\delta\phi^k(\boldsymbol{W})$ to be the sum of the first $k$ eigenvalues.

$$\delta\phi(\boldsymbol{W}) = \sum_{i=1}^{n} \lambda_i \quad \text{and} \quad \delta\phi^k(\boldsymbol{W}) := \sum_{i=1}^{k} \lambda_i. \tag{3}$$

Eq. (3) captures an intuitive notion of loss change in the top-$k$ directions of maximum change in loss. If the update $\delta\boldsymbol{W}$ is aligned with the gradient, i.e., $\boldsymbol{G} \propto \delta\boldsymbol{W}$, then $\delta\phi^k(\boldsymbol{W})$ is the sum of the top-$k$ singular values of $\boldsymbol{G}$. We can add the stepwise changes over time and parameters[1] to compute,

$$\phi^k(\boldsymbol{\omega}) := \sum_{\ell \in [L]} \sum_{t=0}^{T-1} \delta\phi^k\left(\boldsymbol{W}_t^{(\ell)}\right). \tag{4}$$

For some parameters we use a "row version" of $\delta\phi^k$ which uses $\mathcal{S}(\boldsymbol{G}^\top, \delta\boldsymbol{W}^\top)$ instead of $\mathcal{S}(\boldsymbol{G}, \delta\boldsymbol{W})$. Then the index $k$ can span $[n]$ for all layers so we may employ the same $k$ for each layer in Eq. (4).

**Decomposition-aware HP Transfer.** We can now make our earlier intuitions in Hypothesis 1 more precise using this stepwise decomposition. Assume we fix a training procedure $\mathcal{A}$ (see Definition 3). Then we can define the width-$n$ trajectory trained with HPs $\boldsymbol{\nu}$ and scaling $\boldsymbol{\gamma}$ to be the trajectory $\boldsymbol{\omega}_n(\boldsymbol{\nu}, \boldsymbol{\gamma})$ obtained from executing the training procedure $\mathcal{A}$ with hyperparameters $\mathcal{H}_n(\boldsymbol{\nu}, \boldsymbol{\gamma})$. If we consider a fixed scaling $\boldsymbol{\gamma}$ and an HP-dependent *truncation function* $\kappa$ which outputs an index $\kappa(\boldsymbol{\nu}) \in [n]$ given HPs $\boldsymbol{\nu}$, we can abbreviate

$$\phi_n(\boldsymbol{\nu}) := \phi(\boldsymbol{\omega}_n(\boldsymbol{\nu}, \boldsymbol{\gamma})), \ \phi_n^\kappa(\boldsymbol{\nu}) := \phi^{\kappa(\boldsymbol{\nu})}(\boldsymbol{\omega}_n(\boldsymbol{\nu}, \boldsymbol{\gamma}))$$

$$\boldsymbol{\nu}^\star(n) := \arg\min_{\boldsymbol{\nu}} \phi_n(\boldsymbol{\nu}), \ \boldsymbol{\nu}_\kappa^\star(n) := \arg\min_{\boldsymbol{\nu}} \phi_n^{\kappa(\boldsymbol{\nu})}(\boldsymbol{\nu}). \tag{5}$$

---

[1] We will only decompose matrix parameters and use the full inner-product for vector parameters.

Table 1: Notation for the linearized trajectory decomposition.

| Symbol | Definition | Ref. |
|---|---|---|
| $\delta\phi(\boldsymbol{w}_t)$ | Linearized loss change at step $t$. | (1) |
| $\phi(\boldsymbol{\omega})$ | Total sum of $\delta\phi(\boldsymbol{w}_t)$ over trajectory. | (1) |
| $\delta\phi^k(\boldsymbol{W}_t^{(\ell)})$ | Sum of top-$k$ components in layer $\ell$ at time $t$. | (3) |
| $\phi^k(\boldsymbol{\omega})$ | Sum of top-$k$ components over layers and time. | (4) |
| $\phi_n(\boldsymbol{\nu})$ | Loss curve over HPs $\boldsymbol{\nu}$ for width $n$. | (5) |
| $\kappa(\boldsymbol{\nu})$ | Truncation function maps HPs $\boldsymbol{\nu}$ to index in $[n]$. | (5) |
| $\phi_n^\kappa(\boldsymbol{\nu})$ | HP adaptive top-$\kappa$ loss curve $\phi_n^{\kappa(\boldsymbol{\nu})}(\boldsymbol{\nu})$. | (5) |

We will refer to $\phi_n^\kappa$ as the **top-$\kappa$ loss curve** and $\phi_n^{-\kappa} := \phi_n - \phi_n^\kappa$ as the **residual loss curve**. For convenience, we summarize the notation for our trajectory decomposition in Table 1.

Based on this decomposition, we have the following conceptual explanation of fast transfer: if for an appropriately chosen sequence $\kappa_n$ the following are simultaneously true for large enough $n$,

- **Top-$\kappa$ strong convexity:** The top-$\kappa_n$ losses $\phi_n^{\kappa_n}$ and $\phi_\infty^{\kappa_n}$ are locally strongly-convex.
- **Top-$\kappa$ invariance:** The top-$\kappa_n$ loss converges rapidly so $\phi_n^{\kappa_n} \approx \phi_\infty^{\kappa_n}$ and $\boldsymbol{\nu}_{\kappa_n}^\star(n) \approx \boldsymbol{\nu}_{\kappa_n}^\star(\infty)$.
- **Residual Flatness:** Residuals $\phi_n^{-\kappa_n}$, $\phi_\infty^{-\kappa_n}$ are "flat"; hence $\boldsymbol{\nu}_{\kappa_n}^\star(n) \approx \boldsymbol{\nu}^\star(n)$, $\boldsymbol{\nu}_{\kappa_n}^\star(\infty) \approx \boldsymbol{\nu}^\star(\infty)$.

it follows that $\boldsymbol{\nu}^\star(n) \approx \boldsymbol{\nu}^\star(\infty)$, i.e., the optimal HP remains stable across widths $n$ (see Appendix C.5 for details). This is precisely the mechanism posited in Hypothesis 1: the top-$\kappa_n$ loss optimizer stabilizes quickly as width grows and the residual loss is too flat to significantly shift the optimum of the full loss. Due to the prohibitive computational cost to accurately estimate the scaling of the above quantities in realistic settings, in this work we present *qualitative* evidence that the above conditions hold empirically (hence the notation "$\approx$"). We leave a more quantitative investigation to future work.

**Selecting the Truncation Function.** Selecting $\kappa$ involves a trade-off: we must retain enough components to ensure the top-$\kappa$ minimizer approximates the true optimal hyperparameters (requiring larger $k$), while excluding width-sensitive tail components (requiring smaller $k$). In Appendix C.5, we formalize this by defining a quantity $\mathcal{J}_n(\kappa)$ which upper bounds the HP gap $b_n$ using quantitative measures of top-$\kappa$ invariance and residual flatness. By choosing a truncation $\kappa_n^\star$ that minimizes $\mathcal{J}_n(\kappa)$, we balance these properties to obtain the tightest upper bound on $b_n$. Since directly optimizing $\mathcal{J}_n$ is intractable, we instead optimize a proxy objective $\mathcal{J}_{\text{proxy}}(\kappa)$ to obtain a truncation $\hat{\kappa}(n)$ via Algorithm 1 (Appendix D). We validate empirically that $\hat{\kappa}(n)$ achieves the desired qualitative properties.

## 4.2 EXPERIMENTAL RESULTS

We train a Llama-style transformer [TLI+23] using the Adam optimizer [KB14] with a warmup-stable-decay (WSD) learning rate schedule [HTH+24] on WikiText-103 [MXBS16] using $\mu$P, and sweep the peak learning rate as shown in Figure 6; detailed setup can be found in Appendix E.2.1.

In Appendix A.1 we repeat the same setup but with Muon and observe that transfer is less stable and top-$k$ invariance is weaker. In Appendix A.2 we use CIFAR-10 training to interpret what the top and tail components are capturing by introducing a sample-wise version of the top-$k$ decomposition; we use this refined decomposition to connect the quality of transfer to the "hardness" of the samples.

**Fast Transfer and Linearization Faithfulness.** As we can see from Figures 6 and 7, the EMA loss and the linearized loss $\phi$ are nearly indistinguishable, indicating that the EMA trajectory is sufficiently smooth. From Figure 7 we see that smoothing does not degrade the final loss. Our setting also clearly exhibits fast transfer since the optimal learning rate is converging rapidly with the width $n$ (see Fig. 6a) while the reducible loss improves more slowly, converging at a rate of $n^{-0.52}$ (see Fig. 6b). Using the optimal learning rate obtained at width $n = 128$ for larger widths is essentially optimal, as indicated by the overlapping curves in Figure 6b. We now further probe the optimization and scaling dynamics in this fast transfer setting through the lens of our decomposition.

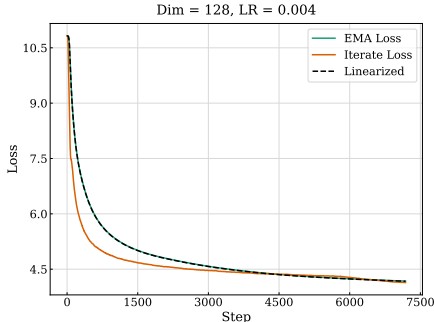

Figure 7: Transformer training on WikiText-103. Linearized and EMA losses are identical through training and match the final iterate loss.

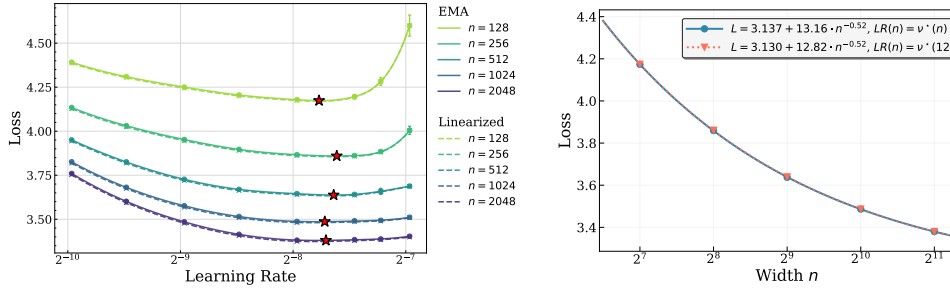

(a) EMA loss (solid) and linearized loss (dashed).

(b) Scaling law for the EMA loss.

Figure 6: Training a 4-layer Llama transformer with Adam optimizer on WikiText-103. **Left:** EMA and linearized losses nearly coincide, indicating small linearization error. **Right:** EMA loss for each width $n$ for two HP choices: the width-dependent optimal learning rate $\nu^\star(n)$ [blue dots] and the width-128 optimal learning rate $\nu^\star(128)$ [orange triangles]. Overlapping curves indicate perfect transfer across widths.

**Decomposition Over Time.** Figure 1 displays the top-$k$ loss with $k = 60$ and residual loss over training, across multiple widths, at a fixed learning rate. We see that throughout training, the top-$k$ loss is nearly width-invariant and accounts for the majority of loss reduction. This indicates that the bulk of the improvement due to optimization comes from a low-dimensional subspace, and the benefit of width mostly comes from improving the residual loss. Accordingly, the "width-dependent" learning largely occurs in the tail components (and their contribution increases later in training).

**Decomposition Across Widths.** To evaluate our fast transfer conjecture, we apply our loss decomposition across widths and compute $\kappa_n = \hat{\kappa}(n)$ using Algorithm 1. We use the largest width $n_{\max} = 2048$ as an infinite-width proxy, and consider transfer from finite widths $n < n_{\max}$ to this proxy. In the right panel of Figure 8, we see that the top-$\kappa_n$ loss curves nearly overlap across learning rates, i.e., $\phi_n^{\kappa_n} \approx \phi_\infty^{\kappa_n}$, despite the large gap in the total losses $\phi_n$ and $\phi_\infty$ shown in the left panel. Furthermore, the minimizers of the total loss are largely determined by the minimizers of the respective top-$\kappa_n$ loss, in the sense that both $\boldsymbol{\nu}_{\kappa_n}^\star(n) \approx \boldsymbol{\nu}^\star(n)$ and $\boldsymbol{\nu}_{\kappa_n}^\star(\infty) \approx \boldsymbol{\nu}^\star(\infty)$ (see Eq. (5)). In Figure 9 we plot the corresponding residuals and see that they are flatter than the top-$\kappa_n$ losses in a neighborhood of the corresponding top-$\kappa_n$ minimizer. As a result, the residuals contribute less to the determination of the overall optimal learning rate.

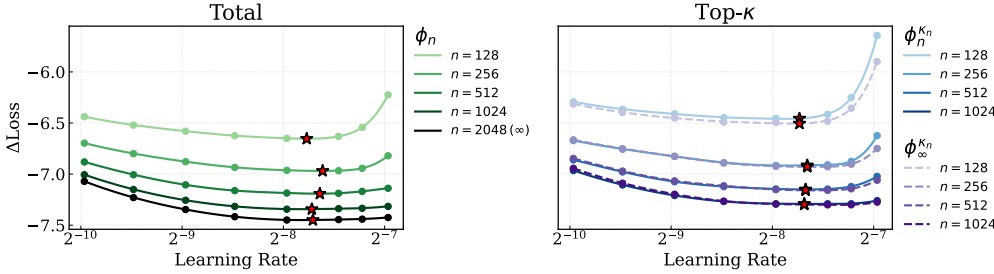

Figure 8: **Left:** Total loss curves $\phi_n$ across widths. **Right:** Top-$\kappa_n$ loss curve pairs $\phi_n^{\kappa_n}$ (blue dashed) and $\phi_\infty^{\kappa_n}$ (purple dashed). The top-$\kappa_n$ pairs nearly overlap, with minimizers close to those of corresponding total losses.

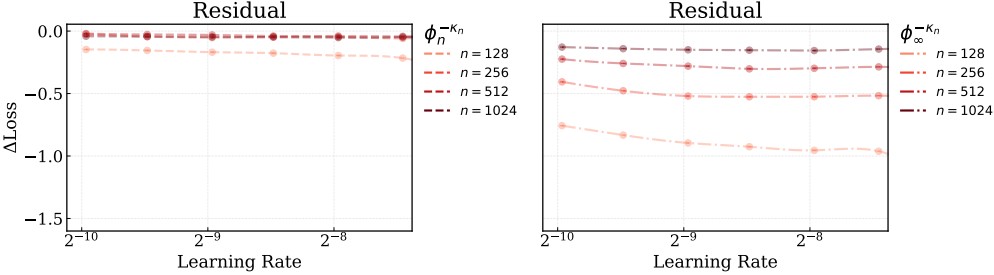

Figure 9: Residual losses are flat around the top-$\kappa_n$ minimizers, indicating less sensitivity to the learning rate.

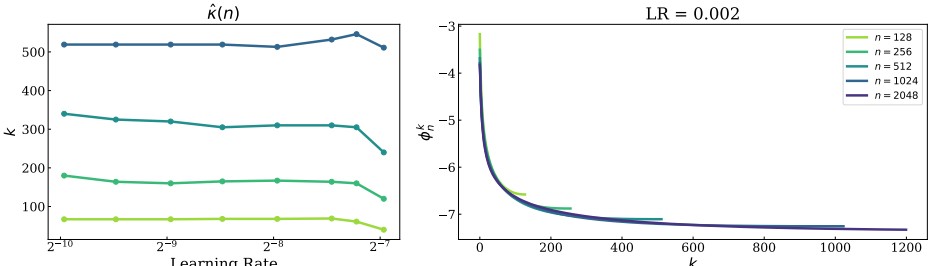

Figure 10: **Left:** Computed values of $\hat{\kappa}(n)$ using Algorithm 1. **Right:** Top-$k$ losses $\phi_n^k$ for LR = 0.002, which descend rapidly with $k$ and overlap across widths over an intermediate range where top-$k$ invariance holds.

**Top-$k$ Profile.** To further interpret the decomposition and the computed values $\hat{\kappa}(n)$, we now examine Figure 10. The left panel shows the computed values of $\hat{\kappa}(n)$, which are approximately constant across learning rates and grow sublinearly with $n$. The right panel plots $\phi_n^k$ for a fixed learning rate as a function of $k$. The top-$k$ loss varies smoothly with $k$, dropping rapidly at first and then flattening out. Notably, we can see that $\hat{\kappa}(n)$ is chosen roughly at the index $k$ where the curve $\phi_n^k$ "peels off" from the $\phi_\infty^k$ curve. This index marks the transition where increasing $k$ starts to include width-sensitive directions and balances the tradeoff discussed at the end of Section 4.1. We see in the figure that this transition point increases with $n$ since the finite-width model shares more converged components with the infinite-width proxy. Consequently, $\hat{\kappa}(n)$ increases with $n$, and the residual $\phi_\infty^{-\kappa_n}$ shown in the right panel of Figure 9 decreases in magnitude. This differs from Figure 1, where we fixed $k = 60$ to isolate a width-invariant index and the residual magnitude grew with $n$. Compared to the infinite-width residual, the finite-width residual $\phi_n^{-\kappa_n}$ is smaller since there are fewer tail components, hence the curves in the left panel of Figure 9 are closer to zero.

Overall, we can qualitatively see how our decomposition can account for fast transfer in the sense described in Section 4, even when the convergence of the loss itself is much slower. The provides concrete evidence for our central hypothesis that there is a low-dimensional projection of the trajectory which remains nearly invariant across width and is responsible for deciding the learning rate. In Appendix E.4.1, we observe similar fast-transfer behavior for GPT-2 architecture trained on FineWeb [PKL+24] using Adam.

## 5 CONCLUSION

This work introduces a novel conceptual framework to reason about hyperparameter transfer and its underlying mechanisms. We posit that a basic form of HP transfer, which we refer to as *weak transfer*, can hold generically as a consequence of the asymptotics of loss and hyperparameter scaling. In synthetic settings, we demonstrate that this asymptotic condition alone does not imply the substantial computational benefit of transfer often observed in practice. We conjecture that *fast & useful transfer* instead requires non-trivial low-dimensional structure in the optimization dynamics. We make this idea concrete by introducing a decomposition of the dynamics based on a linearization of an EMA-smoothed training trajectory. This decomposition provides an operational way to describe the relevant low-dimensional structure and motivates an empirically testable sufficient condition for useful transfer. Our experiments suggest that this structure appears in practice for common optimizers such as SGD and Adam, and that it may underlie the empirical success of hyperparameter transfer across scales. We hope this perspective motivates further work on identifying when efficient transfer should be expected and on developing a deeper understanding of optimization dynamics across scale.

### ACKNOWLEDGMENT

The authors thank Blake Bordelon, Lénaïc Chizat, Jeremy Cohen, Soufiane Hayou, Jason D. Lee, Yan Shuo Tan, Atlas Wang, and Greg Yang for discussion and feedback. The symbolic computation in Appendix C.4 was assisted by GPT5-Pro.

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

CONTENTS

# A  ADDITIONAL FINDINGS ON TOP-$k$ DECOMPOSITION

## A.1  LLAMA WITH MUON

We repeat the same experiments in Section 4.2 but using the recently popularized Muon optimizer [JJB+24], which updates the model weights via the orthogonalized momentum gradients (see Appendix E.2.3 for details). Figure 11a shows that the optimal learning rate shift is more pronounced with Muon than with Adam (see Figure 6a). This larger shift leads to suboptimal performance at larger widths when using transferred hyperparameters, as demonstrated in Figure 11b: the scaling law using the transferred hyperparameter diverges noticeably from the one using the optimal hyperparameters. While this transfer suboptimality is offset by the improved overall performance of Muon, it is still an interesting example of "imperfect" transfer, and our decomposition reveals distinctive properties of Muon's learning dynamics.

Figure 12 reveals that top-$k$ invariance holds only for small $k$ (up to $k_0 \approx 10$). This is in sharp contrast with the decomposition for Adam (Figure 10), in which we observe approximately invariant $\phi_n^k$ (up to $k \approx 60$) that explains a large fraction of the loss reduction. For $k > k_0$, we clearly see that $\phi_n^k$ increases with $n$, even though $\phi_n$ decreases with $n$. This indicates that under our layer-wise decomposition, Muon is "spreading" the loss decrease over more directions: each direction contributes less, but the cumulative decrease over all directions is larger since $\phi_n$ decreases with $n$. Intuitively speaking, this lack of low-dimensional invariance is connected to the "whitening" step in Muon which increases the effective rank of the gradient [FAL25; DD25]. Consequently, the low-rank

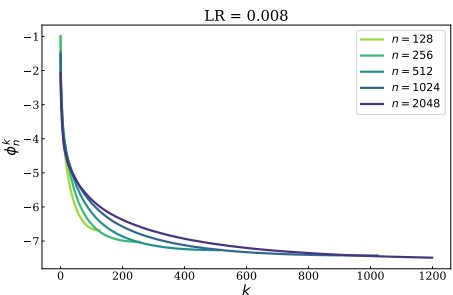

LR = 0.008

Figure 12: Top-$k$ losses $\phi_n^k$ for Muon. The curves flatten out slowly with $k$, especially for large $n$, showing top-$k$ invariance holds only for a narrow range of $k$ and the residual is significant.

structure in Adam and SGD updates that enables fast transfer may not be present in full-matrix preconditioned updates.

In Appendix E.4.2, we report the learning rate transfer of Muon in a different problem setting: training GPT-2 on the FineWeb dataset. There again we find that Muon displays a less stable optimal learning rate across widths, but due to the flatness of the overall loss, the transfer performance is still nearly optimal. Interestingly, in Figure 32 we observe that the decomposition looks qualitatively different for different learning rates. In particular, while the small learning rate decomposition resembles that in our earlier Muon experiment (Figure 12), at near-optimal learning rates we instead see more pronounced top-$k$ invariance that is closer to the Adam decomposition (Figure 10). We speculate that this can happen when the dynamics of the layers trained with AdamW (e.g., the input and output layers, see Appendix B.2) tend to dominate at certain learning rates, but we leave careful verification of this to future work. In any case, this example highlights that the correct notion of invariance for Muon is subtle and can be hyperparameter dependent.

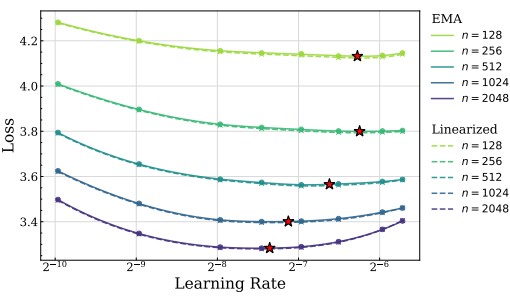

(a) EMA loss (solid) and linearized loss (dashed).

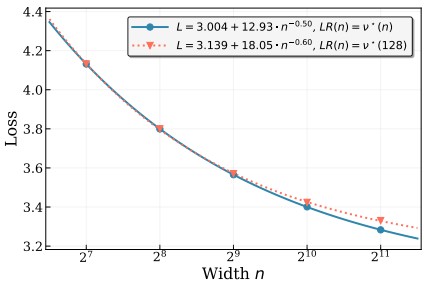

(b) Scaling law for the EMA loss.

Figure 11: Same model and dataset as Fig. 6, but trained with Muon. **Left:** EMA and linearized losses coincide. **Right:** EMA loss versus width $n$ for the learning rate choices $\nu^\star(n)$ [blue dots] and $\nu^\star(128)$ [orange triangles]. At large $n$, using $\nu^\star(128)$ becomes suboptimal, indicating imperfect transfer.

**Experiment with Muon Variants.**    To investigate the effect of whitening on learning rate transfer and the decomposition in a more controlled manner, in Appendix B.3 we perform the same experiment using the Dion optimizer [AXA$^+$25], which has a rank hyperparameter controlling the dimension of the whitened subspace. If we choose the Dion rank to be small, we expect a higher degree of invariance and less influence of the residual, which should lead to improved transfer. In Figure 24, we see that setting the rank equal to $\min(n/2, 128)$ indeed greatly improves transfer compared to using a rank of $n/2$, which closely resembles the Muon results (Figure 11). Perhaps surprisingly, although Dion with bounded rank does exhibit more stable transfer, the resulting top-$k$ decomposition (Figure 26a) is more Adam-like, but still looks qualitatively similar to that for Muon. We speculate that there exists a different notion of invariance[2] for optimizers like Muon and Dion, obtained by modifying our proposed decomposition, that leads more cleanly to a qualitative picture similar to what we see for SGD and Adam; for such suitable notion of invariance, we expect the corresponding residual loss will be flatter for Dion with bounded rank compared to Muon.

## A.2   MLP with SGD

For a setting with a new data modality and architecture, we now consider learning rate transfer in two-layer MLPs trained on CIFAR-10 using momentum SGD. Additional details can be found in Appendix E.5. This setting will have the benefit of being lightweight enough to support a detailed analysis and interpretation of our decomposition from a *data-centric* point of view (similar in spirit to [ZGKS21; KGG$^+$22; IPE$^+$22]). In Figure 13 we can see that our linearization described in Section 4.1 is accurate and the optimal HP is stable across width (though transfer is "imperfect" compared to Figure 6). We also observe that the benefit of width is less pronounced compared to the language setting and that loss convergence occurs at a faster rate of approximately $n^{-0.77}$. This aligns with the intuition that CIFAR-10 is a "simpler" task; in Appendix E.5 we further support this intuition through the top-$k$ decomposition (Fig. 33).

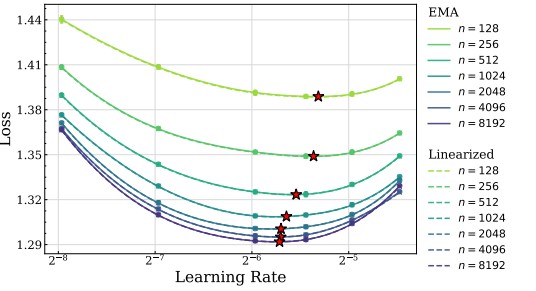
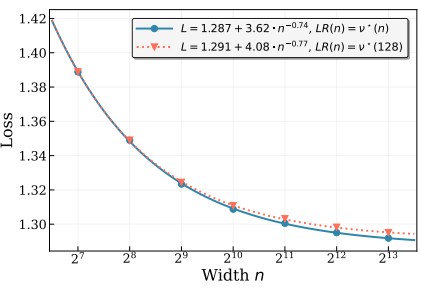

(a) The final EMA loss (solid) and final linearized.

(b) Scaling law for the EMA loss.

Figure 13: Two-layer MLP trained on CIFAR-10 using SGD. We observe reliable but imperfect transfer. **Left:** The EMA and linearized losses coincide. **Right:** EMA loss across widths using the width-dependent optimal learning rate $\nu^\star(n)$ [blue dots] and the fixed width-128 choice $\nu^\star(128)$ [orange triangles].

**Sample-wise Decomposition.**    We have previously seen that the top-$k$ components account for the majority of loss decrease, while the tail components account for the improved learning at larger widths. However, this picture does not shed light on what structures the top vs. tail components are actually learning. To gain a better qualitative understanding, we now apply our decomposition at a sample-wise level and examine how different components affect the loss on each example. We conjecture the following structure:

- **Easy Examples:** Examples that are almost entirely learned by the *top components* are "easy" examples. Loss on these examples will show fast transfer and essentially decide the optimal HPs.

- **Hard Examples:** Examples that rely on the *tail components* are "hard" examples. These examples are learned differently across widths because tail components are not width-invariant, hence slowing transfer.

---

[2]For example, it could make sense to consider invariance for the *rescaled* top-$k$ loss $\tilde{\phi}_n^k := \phi_n^{m_n(k)}$ where $m_n(k) = \alpha k n^\beta$ for some $\alpha > 0$, $\beta \in (0,1)$ and $k \in [n^{1-\beta}/\alpha]$, since this would capture an increasing number of directions with $n$.

**Mean Component Index.**    To make these notions precise, we now introduce a per-example statistic, the *Mean Component Index* (MCI), which quantifies how much of the loss change for an example occurs in top versus tail components. Examples with small MCI correspond to the "easy" examples above and those with high MCI correspond to the "hard" examples. Let us recall the setup in Section 4.1. For simplicity, let us consider a single layer $W$ with gradient $G$ and update $\delta W$ at time $t$, since everything will extend to multiple layers by summing over each layer. As before, let $\mathcal{S}(G, \delta W)$ from Eq. (2) be the alignment matrix between $G$ and $\delta W$ with eigenvalues $\lambda_1, \ldots, \lambda_n$ ordered such that $|\lambda_1| \geq \cdots \geq |\lambda_n|$ and corresponding eigenvectors $u_1, \ldots, u_n$. The eigenvectors $u_j$ define the component directions in our decomposition. If we have $P$ data points and the loss $\mathcal{L}$ is the average over pointwise losses $\ell_i$ with gradients $G_i$, then $G$ is the average over $G_i$ and we can decompose $\delta\phi(W) = \langle G, \delta W \rangle$ into

$$\delta\phi(W) = \langle G, \delta W \rangle = \frac{1}{P}\sum_{i=1}^{P} \langle G_i, \delta W \rangle = \frac{1}{P}\sum_{i=1}^{P}\sum_{j=1}^{n} u_j^\top G_i^\top \delta W u_j = \frac{1}{P}\sum_{i=1}^{P}\sum_{j=1}^{n} (\delta\psi)_{ij},$$

where we define $(\delta\psi)_{ij} := u_j^\top G_i^\top \delta W u_j$ to be the instantaneous linearized loss change of sample $i$ in the $j$ component from updating parameter $W$. By summing $(\delta\psi)_{ij}$ over time steps $t$ and matrix layers we can define $\psi_{ij}$ to be the linearized loss change of sample $i$ in the $j$ component over the entire trajectory. The mean component index $\mathrm{MCI}_i \in [1, n]$ for example $i$ is defined as the quantity

$$\mathrm{MCI}_i := \sum_{j=1}^{n} j p_j, \quad p_j := \frac{|\psi_{ij}|}{\sum_{k=1}^{n} |\psi_{ik}|} \quad \text{and} \quad \psi_{ij} := \sum_{\ell \in [L]} \sum_{t=0}^{T-1} (\delta\psi)_{ij}[W_t^{(\ell)}]. \tag{6}$$

We use $\mathrm{MCI}_i$ as a scalar index placing example $i$ along the easy-to-hard spectrum described above.

**Empirical Findings.**    In Figure 14a we show the distribution of $\mathrm{MCI}_i$ for a network of width $n = 128$. The average $\mathrm{MCI}_i$ is about 12.9, which is much smaller than $n$, and the distribution is left-skewed. This indicates that most samples are learned mainly through the top components, consistent with Figures 33 and 35.

In Figure 15 we visualize the four examples with the smallest and largest MCI. Observe that examples with smallest MCI shown in Figure 15a are visually simple (top panel) and are essentially learned by the first component (bottom panel) in our decomposition (6), whereas the large MCI examples shown in Figure 15b are more complex and require non-trivial contributions from tail components (i.e., $j > 10$).

Figure 14b quantifies how similar the rankings of $\mathrm{MCI}_i$ over samples $i$ are across widths and random seeds. For widths $n_1, n_2$ and random seeds $s_1, s_2$, we take the top or bottom 5% of samples ranked by MCI for each width and seed pair, giving us two subsets $S_1$ and $S_2$ of $[P]$. For a given pair of widths we compute the fraction of overlap $|S_1 \cap S_2|/0.05P$ and average over pairs of seeds $s_1$ and $s_2$, ignoring $s_1 = s_2$ if $n_1 = n_2$ since this would be trivially equal to one. We observe less agreement when $\min(n_1, n_2)$ is smaller and we restrict to samples with higher MCI, suggesting that easy (low-MCI) examples are learned more consistently at different widths and random seeds than hard (high-MCI) examples.

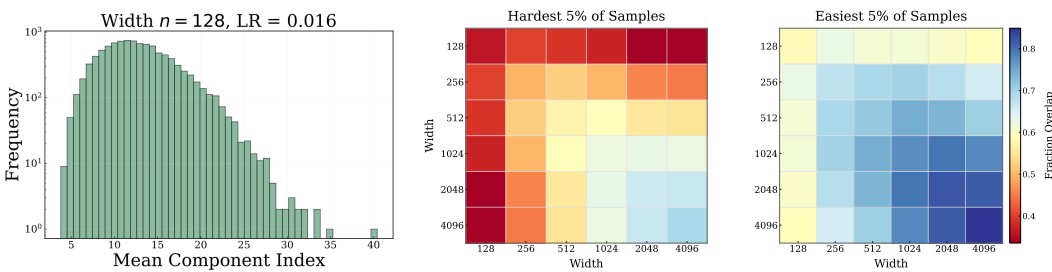

(a) Histogram of MCI (CIFAR-10).          (b) Consistency of top 5% hardest and easiest examples.

Figure 14: **Left:** MCI computed on CIFAR-10 for $n = 128$ and learning rate 0.016. The distribution is left-skewed with a small number of large outliers, indicating that most examples are "easy". **Right:** For each pair of widths and random seeds, we compare the sets of top or bottom 5% of samples ranked by $\mathrm{MCI}_i$ and plot the average fraction of shared samples. Diagonal values need not be 1 since entries compare different seeds at the same width. Rankings are more stable for low-MCI (easy) examples and larger widths, and less stable for high-MCI examples and smaller widths.

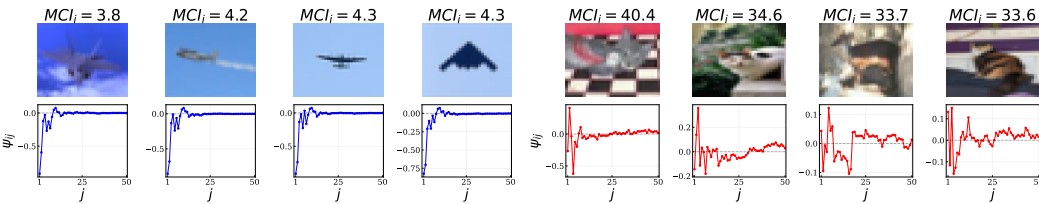

(a) Lowest Mean Component Index ("easy" samples).  (b) Highest Mean Component Index ("hard" samples).

Figure 15: The five examples in CIFAR-10 validation set with lowest (left) and highest (right) MCI for a two-layer MLP of width $n = 128$ trained with SGD using LR = 0.016. **Top row:** We visualize the samples and the corresponding MCI. **Bottom row:** We plot the $\psi_{ij}$ values for the first 50 components $j$ (see Eq. (6)). Note that the low-MCI examples turn out to be simple images from the airplane class which are learned early in training using simple features, while the high-MCI images are complex examples from different classes which are not learned well by the model.

Due to the consistency across scale observed in Figure 15a, we intuitively expect the optimal learning rate for these easy examples to be stable and transferrable. Hence in Figure 16 we probe the effect of easy versus hard examples on learning rate transfer. Up to this point, all losses have been computed on the full validation set. To isolate the effect of individual data points, we now evaluate learning-rate sweeps on subsets of the validation set consisting only of easy or hard examples, based on the previously computed MCI values from training a single model of with $n = 128$. On the easy subset (lowest 25% MCI) we see near-perfect transfer (Figures 16a, 16b), and the optimal learning rate is very close to the one obtained on the full validation set reported in Figure 13; this suggests that the optimal HP on easy examples – learned by the top components and remains stable across width – approximately decides the optimal HP for the full loss. In contrast, on the hard subset (highest 5% MCI) we see a clear leftward shift in the optimal learning rate as width increases and noticeably worse transfer (Figures 16c, 16d). This illustrates a potential failure mode of HP transfer in practice and shows how our sample-based decomposition can help reveal such failures: if the downstream evaluation metric $\phi$ is "out-of-distribution" and concentrated on high-MCI examples, then HPs transferred from a small model can yield suboptimal performance. We leave further investigation of this sample-wise decomposition, for example in language model settings, as future work.

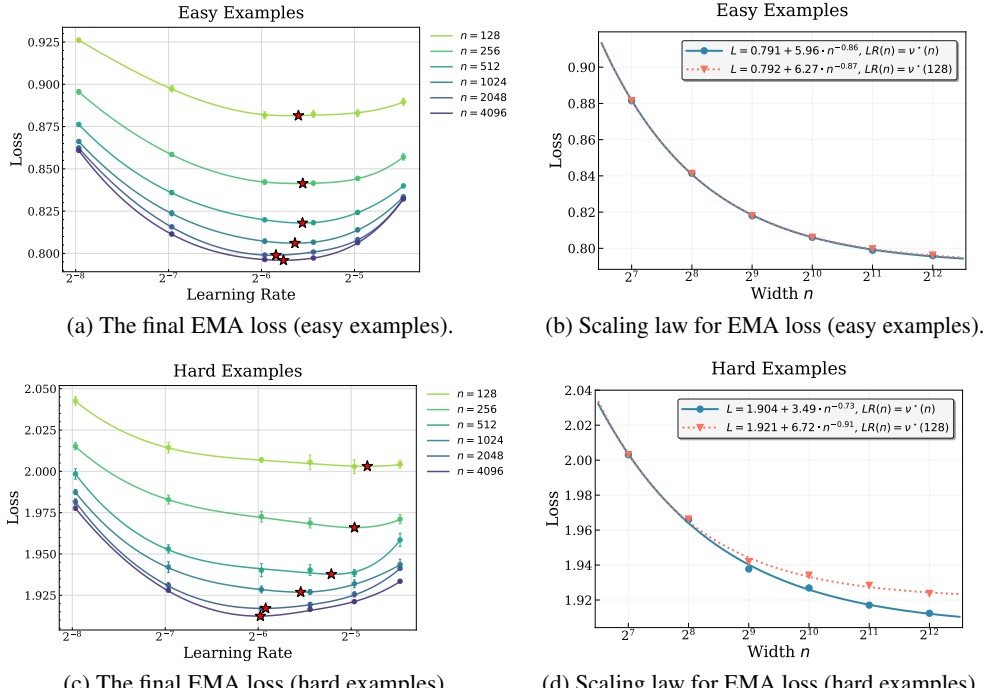

(a) The final EMA loss (easy examples).  (b) Scaling law for EMA loss (easy examples).

(c) The final EMA loss (hard examples).  (d) Scaling law for EMA loss (hard examples).

Figure 16: EMA loss on the 25% easiest **(Top)** and 5% hardest **(Bottom)** examples in the CIFAR-10 validation set, based on MCI computed on the width $n = 128$, LR = 0.016 model. Observe that for EMA loss on the easy subset, the optimal learning rates are well-aligned across widths and match the (large width) optimal HP on the full validation set (Figure 13a), whereas the optimal learning rates for the hard subset shift significantly and yields suboptimal transfer.

# B BACKGROUND

**Function Class Regularity** Let $\mathcal{X}$ be a compact metric space, and let $C(\mathcal{X})$ denote the space of real-valued continuous functions on $\mathcal{X}$, equipped with the uniform norm:

$$\|f\|_{\text{sup}} := \sup_{\boldsymbol{\nu} \in \mathcal{X}} |f(\boldsymbol{\nu})|.$$

We say a collection $\mathscr{F} \subset C(\mathcal{X})$ is:

- *uniformly bounded* if $\sup_{f \in \mathscr{F}} \|f\|_{\text{sup}} \leq K$ for some $K < \infty$,

- *uniformly equicontinuous* if for every $\varepsilon > 0$ there exists $\delta > 0$ such that

$$\|\boldsymbol{\nu} - \boldsymbol{\nu}'\| < \delta \implies |f(\boldsymbol{\nu}) - f(\boldsymbol{\nu}')| < \varepsilon \quad \text{for all } f \in \mathscr{F}.$$

We denote by $C^k(\mathcal{X})$ the space of $k$-times continuously differentiable functions. For $f \in C^k(\mathcal{X})$, the $k$-th derivative is written $f^{(k)}$. We define $f^{(0)} \equiv f$. In multivariate settings, this refers to the $k$-th total derivative.

**Theorem 4** (Arzelà–Ascoli). *Any uniformly bounded and uniformly equicontinuous collection $\mathscr{F} \subset C(\mathcal{X})$ is relatively compact in the uniform norm topology.*

**Proposition 5.** *Let $\{f_n\} \subset C^1(\mathcal{X})$ such that $f_n' \to g$ uniformly and $f_n(\boldsymbol{\nu}_0) \to L$ for some $\boldsymbol{\nu}_0 \in \mathcal{X}$ and $L \in \mathbb{R}$. Then $f_n \to f$ uniformly for some $f \in C^1(\mathcal{X})$, and $f' = g$.*

**Proposition 6.** *If $\{f_n\} \subset C^1(\mathcal{X})$ and the derivatives $f_n'$ are uniformly bounded, then $\{f_n\}$ is uniformly equicontinuous.*

## B.1 SCALING LIMITS AND TENSOR PROGRAMS

In this section we will recall some simplified background from [YH21; YL23]. For concreteness, we will fix the architecture to a $L$-hidden layer MLP, but all statements can be extended to a much more generic architectures (see Section 2.9.1 in [YL23]). An $L$-hidden layer MLP of width $n$ with nonlinearity $\phi : \mathbb{R} \to \mathbb{R}$ and no biases is parameterized by weight matrices $\boldsymbol{W}^1 \in \mathbb{R}^{n \times d}$, $\boldsymbol{W}^2, \ldots, \boldsymbol{W}^L \in \mathbb{R}^{n \times n}$, and $\boldsymbol{W}^{L+1} \in \mathbb{R}^{1 \times n}$. On an input $\boldsymbol{x} \in \mathbb{R}^d$, the network computes

$$\boldsymbol{h}^\ell(\boldsymbol{x}) = \boldsymbol{W}^\ell \boldsymbol{z}^\ell(\boldsymbol{x}) \in \mathbb{R}^n, \ \ \boldsymbol{z}^\ell(\boldsymbol{x}) = \phi(\boldsymbol{h}^\ell(\boldsymbol{x})) \in \mathbb{R}^n, \ \ \text{for } \ell = 1, \ldots, L, \tag{7}$$

and the output is $f(\boldsymbol{x}) = \boldsymbol{W}^{L+1} \boldsymbol{z}^L(\boldsymbol{x}) \in \mathbb{R}$. Given $N$ inputs $\boldsymbol{x}_1, \ldots, \boldsymbol{x}_N$, we will abbreviate

$$\boldsymbol{h}^\ell := [\boldsymbol{h}^\ell(\boldsymbol{x}_1) \mid \cdots \mid \boldsymbol{h}^\ell(\boldsymbol{x}_N)] \in \mathbb{R}^{n \times N},$$
$$\boldsymbol{z}^\ell := [\boldsymbol{z}^\ell(\boldsymbol{x}_1) \mid \cdots \mid \boldsymbol{z}^\ell(\boldsymbol{x}_N)] \in \mathbb{R}^{n \times N},$$
$$\boldsymbol{f} := (f(\boldsymbol{x}_1), \ldots, f(\boldsymbol{x}_N)) \in \mathbb{R}^N.$$

**abc-parameterization** Assume that we train the network using SGD. We recall the definition of abc-parameterization from [YH21] (see [GSJW20; CN24] for similar findings). An abc-parameterization is a width-aware HP scaling specified by a set of HPs $\boldsymbol{\nu} = \{\alpha_\ell, \sigma_\ell, \eta_\ell\}_{\ell \in [L+1]}$ and HP scaling exponents $\boldsymbol{\gamma} = \{a_\ell, b_\ell, c_\ell\}_{\ell \in [L+1]}$ such that

(a) The weights $\boldsymbol{W}^\ell$ receive a multiplier $\alpha_\ell n^{-a_\ell}$,

(b) We initialize each $W_{\alpha\beta}^\ell \sim \mathcal{N}(0, \sigma_\ell^2 n^{-2b_\ell})$, and

(c) The SGD learning rate in layer $\ell$ is $\eta_\ell n^{-c_\ell}$.

**Asymptotic Notation** Given a sequence $\boldsymbol{x} = \{\boldsymbol{x}(n)\}_{n=1}^\infty$ of random tensors we write $\boldsymbol{x} = \Theta(n^{-a})$ and say that $\boldsymbol{x}$ has coordinates of size $\Theta(n^{-a})$ if there exist constants $A, B > 0$ such that, almost surely for sufficiently large $n$,

$$A \leq \frac{1}{\#\boldsymbol{x}(n)} \sum_\alpha \boldsymbol{x}(n)_\alpha^2 \leq B,$$

where $\#\boldsymbol{x}(n)$ denotes the number of entries in $\boldsymbol{x}(n)$. We use $O(n^{-a})$ and $\Omega(n^{-a})$ similarly.

**Dynamical Dichotomy Theorem.** We recall some definitions from [YL23]. To reflect the network after $t$ steps of training we add a subscript $t$ to the quantities in Eq. (7). We use $\Delta$ to denote a one-step difference of a time-dependent quantity. We say an abc-parameterization is

1. *stable at initialization* if
$$\boldsymbol{h}_0^\ell, \boldsymbol{z}_0^\ell = \Theta(1), \, \forall \ell \in [L], \text{ and } \boldsymbol{f}_0 = O(1).$$

2. *stable during training* if for any time $t \geq 0$ we have for any training routine
$$\Delta \boldsymbol{h}_t^\ell, \Delta \boldsymbol{z}_t^\ell = O(1), \, \forall \ell \in [L], \text{ and } \Delta \boldsymbol{f}_t = O(1).$$

3. *trivial* if for any time $t \geq 1$ and training routine, $\boldsymbol{f}_t - \boldsymbol{f}_0 \to 0$ almost surely as $n \to \infty$. We say the parameterization is *non-trivial* otherwise.

4. is in the *kernel regime* if there exists $\mathcal{K} : \mathbb{R}^N \to \mathbb{R}^N$ such that for every $t \geq 0$ and training routine, as $n \to \infty$,
$$\boldsymbol{f}_{t+1} - \boldsymbol{f}_t - \eta \mathcal{K}(\boldsymbol{f}_t) \to 0.$$

5. is *feature learning* if $\Delta \boldsymbol{z}_t^L = \Omega(1)$ for some training routine and $t \geq 0$.

**Theorem 7** (Dynamical Dichotomy [YH21]). *A nontrivial and stable abc-parametrization either admits feature learning or is in the kernel regime but not both. The kernel regime does not admit feature learning, that is, for any training routine $\Delta \boldsymbol{z}_t^L \to 0$ for all $t \geq 0$.*

The $\mu$P and NTK parameterization are the maximal feature learning and kernel parameterizations respectively. All other such parameterizations can be obtained from one of these by setting some initialization or learning rate to zero (see Section 5.3 in [YH21] for more discussion). For adaptive optimizers such as Adam it is possible to extend the definitions to abcd-parameterizations (see Section 2.2 in [YL23] for more details), for which a similar Dynamical Dichotomy theorem exists.

## B.2 MUON OPTIMIZER

Here we recall the Muon optimizer introduced by [JJB+24]. For transformers typically the embedding and output layers, as well as vector and scalar parameters are trained using Adam. For the remaining matrix parameters, the update is built around the *matrix sign function* $\mathrm{msgn}(\cdot)$, which for a full-rank matrix $\boldsymbol{X}$ with SVD $\boldsymbol{X} = \boldsymbol{U\Sigma V}^\top$ is defined as
$$\mathrm{msgn}(\boldsymbol{X}) = \boldsymbol{UV}^\top.$$

For parameter $\boldsymbol{W} \in \mathbb{R}^{m \times n}$ with gradient $\boldsymbol{G} \in \mathbb{R}^{m \times n}$, at step $t$ we form a momentum matrix
$$\boldsymbol{M}_{t+1} = \beta \boldsymbol{M}_t + (1 - \beta) \boldsymbol{G}_t,$$

with momentum parameter $\beta \in [0, 1)$, and then update the weights by
$$\boldsymbol{W}_{t+1} = \boldsymbol{W}_t - \eta \sqrt{\frac{m}{n}} p(\boldsymbol{M}_t) \approx \boldsymbol{W}_t - \eta \sqrt{\frac{m}{n}} \mathrm{msgn}(\boldsymbol{M}_t),$$

where $\eta$ is the learning rate, the factor $\sqrt{m/n}$ ensures $\mu$P scaling, and $p$ is a composition of low-degree matrix polynomial chosen to approximate the function $\mathrm{msgn}$. This approximation comes from an iterative Newton-Schulz method where the number of iterations gives the number of polynomial compositions in $p$.

## B.3 DION OPTIMIZER

Dion [AXA+25] is an orthonormalization-based optimizer designed to retain the benefits of Muon-style matrix updates while remaining compatible with sharded weight layouts in large-scale LLM training. Instead of applying a Newton–Schulz iteration on each weight matrix, Dion operates on a momentum buffer and uses an amortized power iteration to obtain an orthonormal low-rank update, controlled by a rank hyperparameter and stabilized by error feedback.

For a matrix parameter $\boldsymbol{W} \in \mathbb{R}^{m \times n}$ with gradient $\boldsymbol{G} \in \mathbb{R}^{m \times n}$ at step $t$, Dion maintains a residual momentum matrix $\boldsymbol{M} \in \mathbb{R}^{m \times n}$ and a right factor $\boldsymbol{Q} \in \mathbb{R}^{n \times r}$, where $r$ is the rank hyperparameter. Each step begins by accumulating the new gradient into the residual momentum,
$$\widetilde{\boldsymbol{M}}_{t+1} = \boldsymbol{M}_t + \boldsymbol{G}_t.$$

An approximate leading left subspace is then extracted using the current $\boldsymbol{Q}_t$,

$$\boldsymbol{P}_t = \operatorname{orth}\big(\widetilde{\boldsymbol{M}}_{t+1}\boldsymbol{Q}_t\big), \qquad \boldsymbol{R}_t = \widetilde{\boldsymbol{M}}_{t+1}^{\top}\boldsymbol{P}_t,$$

where $\operatorname{orth}(\cdot)$ returns a matrix with orthonormal columns (e.g. via a QR decomposition). The right factor is updated by column-wise normalization,

$$\boldsymbol{Q}_{t+1}^{(j)} = \frac{\boldsymbol{R}_t^{(j)}}{\|\boldsymbol{R}_t^{(j)}\|_2 + \varepsilon}, \qquad j = 1,\ldots,r,$$

with a small $\varepsilon > 0$ for numerical stability. When $\boldsymbol{Q}_t$ spans the dominant right-singular subspace of $\widetilde{\boldsymbol{M}}_{t+1}$, this amortized power iteration drives $\boldsymbol{P}_t$ and $\boldsymbol{Q}_{t+1}$ toward approximate left and right singular vectors of $\widetilde{\boldsymbol{M}}_{t+1}$, with both factors having unit-norm columns.

The residual momentum is updated with error feedback,

$$\boldsymbol{M}_{t+1} = \widetilde{\boldsymbol{M}}_{t+1} - (1-\mu)\,\boldsymbol{P}_t\boldsymbol{R}_t^{\top},$$

where $\mu \in [0,1)$ controls how aggressively the dominant rank-$r$ component is removed. In the idealized regime where $\boldsymbol{P}_t\boldsymbol{R}_t^{\top}$ captures the leading rank-$r$ part of $\widetilde{\boldsymbol{M}}_{t+1}$, this update geometrically damps that component by a factor $\mu$ while retaining higher-rank and previously truncated directions in $\boldsymbol{M}_{t+1}$. These retained components can subsequently re-enter the leading low-rank subspace, so the residual acts as an error buffer that compensates for the information lost by low-rank truncation.

The weight update Dion uses is

$$\boldsymbol{W}_{t+1} = \boldsymbol{W}_t - \eta\sqrt{\frac{m}{n}}\,\boldsymbol{P}_t\boldsymbol{Q}_{t+1}^{\top},$$

where the learning rate is $\eta$ and the factor $\sqrt{m/n}$ ensures $\mu$P scaling.

## C  THEORETICAL RESULTS

### C.1  WEAK TRANSFER

In the following we provide a minimal set of technical conditions for the concept of HP transfer to be well-defined and align with empirical observations. We say that a function is *locally strongly convex* (LSC) with parameters $\tau, \delta > 0$, if for every minimizer $\boldsymbol{\nu}^\star \in \arg\min f$,

$$|f(\boldsymbol{\nu}) - f(\boldsymbol{\nu}^\star)| \geq \frac{\tau}{2}\|\boldsymbol{\nu} - \boldsymbol{\nu}^\star\|^2, \text{ for all } \boldsymbol{\nu} \text{ such that } \|\boldsymbol{\nu} - \boldsymbol{\nu}^\star\| \leq \delta.$$

Recall that we assume that the HP search takes place over a *search space* $\mathcal{X}$:

$$\mathcal{X} = \prod_{i=1}^{h}[\ell_i, u_i], \ \text{int}(\mathcal{X}) := \prod_{i=1}^{h}(\ell_i, u_i),$$

which is a $h$-dimensional box with bounds $\ell_i < u_i$ and interior $\text{int}(\mathcal{X})$.

**Definition 3** (HP Transfer). *The scaling $\boldsymbol{\gamma}$ admits HP transfer for a training procedure $\mathcal{A}$ and metric $\phi$ over a search space $\mathcal{X}$ if the following hold almost surely*

1. *$\phi_n \in C^2(\mathcal{X})$ is convex and has a unique minimizer $\boldsymbol{\nu}^\star(n)$.*

2. *There exists LSC deterministic $\phi_\infty$ such that $\phi_n \to \phi_\infty$ and $\arg\min \phi_\infty \subseteq \text{int}(\mathcal{X})$.*

3. *The family $\{\phi_n''\}$ is uniformly bounded and uniformly equicontinuous.*

This definition implies the following desirable consequences.

**Proposition 8** (HP Transfer Properties). *In the context of Definition 3, the following are true*

1. *$\phi_\infty \in C^2(\mathcal{X})$ is convex and has a unique minimizer $\boldsymbol{\nu}^\star(\infty)$*

2. *$\phi_n \to \phi_\infty$, $\phi_n' \to \phi_\infty'$, $\phi_n'' \to \phi_\infty''$ uniformly and $\boldsymbol{\nu}^\star(n) \to \boldsymbol{\nu}^\star(\infty)$.*

*Proof.* By Proposition 6, it follows that for all $j \in \{0, 1, 2\}$, $\phi_n^{(j)}$ is uniformly equicontinuous. Since $\phi_n$ has a limit, it must converge uniformly $\phi_\infty$. Now repeatedly using Proposition 5 after passing to subsequences and invoking Arzela-Ascoli (Theorem 4), we see that $\phi_n^{(j)} \to \phi_\infty^{(j)}$ uniformly for $j \in \{1, 2\}$. Now it follows that $\phi_\infty \in C^2(\mathcal{X})$ since the uniform limit of continuous functions is continuous and $\phi_\infty$ is convex since convexity is preserved under pointwise limits. Note that the local strong convexity condition implies that $\phi_\infty$ has a unique minimizer. Now because $\mathcal{X}$ is compact, every subsequence of $\boldsymbol{\nu}^\star(n)$ has a convergent subsequence. By uniform convergence and the continuity of the $\phi_n$ and $\phi_\infty$, it follows that the limit of this subsequence is a minimizer of $\phi_\infty$. By the uniqueness of this minimizer, all the subsequences converge to $\boldsymbol{\nu}^\star(\infty)$ hence $\boldsymbol{\nu}^\star(n) \to \boldsymbol{\nu}^\star(\infty)$. $\square$

This definition gives a minimal set of conditions under which the concept becomes mathematically coherent and aligns with observed empirical behavior. Requiring $\phi_n$ to have a unique minimizer removes ambiguity about which configuration should be transferred across scales. The convexity, smoothness, and equicontinuity conditions provide technical regularity that facilitates analysis and generally hold in practice.

The local strong convexity of $\phi_\infty$ ensures that performance meaningfully degrades away from the optimum. Without this condition, $\phi_\infty$ may be flat near its minimizer, making accurate hyperparameter selection potentially irrelevant in the large-$n$ limit. The assumption that $\boldsymbol{\nu}^\star(\infty)$ lies in the interior of the search space ensures that this optimum remains unchanged under any enlargement of the domain; this condition rules out the $n$-dependent drift of the HPs of interest due to a "suboptimal" scaling – see Appendix B for discussions.

**"Optimal" scaling limit.**  In [YHB+22; YYZH23], the authors remark that a key principle behind hyperparameter transfer is the "optimality" of the scaling. Heuristically if a scaling yields a suboptimal limit, then it cannot exhibit hyperparameter transfer since the HPs $\boldsymbol{\nu}$ need to undergo a $n$-dependent rescaling to "convert" the suboptimal scaling into the optimal scaling. The following proposition formalizes this intuition. The proposition requires $\mathbf{0} \in \mathcal{X}$ to avoid uninteresting cases where the optimal HP is zero which will generally not occur for optimization HPs in normal neural network training.

**Theorem 9.** *Let $\gamma$ be a scaling that exhibits transfer over $\mathcal{X} = [0, u_1] \times \cdots [0, u_k]$ and all $\mathcal{X}'$ containing $\mathcal{X}$. Any other scaling $\gamma' \neq \gamma$ with these properties must satisfy*

$$\min_{\nu \in \mathcal{X}} \phi_\infty(\nu; \gamma) = \min_{\nu' \in \mathcal{X}'} \phi_\infty(\nu'; \gamma').$$

*Proof of Theorem 9.* For brevity define

$$\nu_\star := \arg\min_{\nu \in \mathcal{X}} \phi_\infty(\nu; \gamma) \quad \text{and} \quad \nu'_\star := \arg\min_{\nu' \in \mathcal{X}} \phi_\infty(\nu'; \gamma').$$

For the sake of contradiction assume that $\phi_\infty(\nu_\star; \gamma) > \phi_\infty(\nu'_\star; \gamma')$. For large enough $n$, we will have $\phi_n(\nu_\star; \gamma) > \phi_n(\overline{\nu}_\star; \gamma)$ where $\overline{\nu}_\star = \nu'_\star \odot (n^{\gamma_1 - \gamma'_1}, \ldots, n^{\gamma_k - \gamma'_k})$ and $\nu_\star \neq \overline{\nu}_\star$ which is a contradiction. The case $\phi_\infty(\nu_\star; \gamma) < \phi_\infty(\nu'_\star; \gamma')$ follows analogously. □

The dynamical dichotomy theorem states that all scalings induced by abcd-parameterizations except for $\mu$P lead to optimization degeneracies or non-feature learning behavior. Therefore the proposition implies that if HP transfer is possible with $\mu$P and feature learning is advantageous, then it should be only possible using $\mu$P. Of course, it is still a challenging problem to rigorously characterize when $\mu$P will exhibit transfer and when feature learning is actually advantageous.

## C.2 ASYMPTOTIC RATES

Recall the definitions of the quantities $a_n, b_n, c_n$ from Definition 1. We will need to reason about the following quantity which we call the *uniform loss gap*.

$$\bar{a}_n := \|\phi_n - \phi_\infty\|_{\sup}. \tag{8}$$

Using the local strong convexity assumption, we will be able to directly relate the HP gap with the uniform loss gap. We first prove a convenient lemma which bounds the minimizer displacement in terms of the sup-norm of the perturbation.

**Lemma 10.** *Let $f : \mathcal{X} \to \mathbb{R}$ and $g : \mathcal{X} \to \mathbb{R}$ such that $f$ has a unique minimizer $x_f$ and $g$ has a unique minimizer $x_g$, and $g$ is $\tau$ strongly-convex*

$$g(x) - g(x_g) \geq \frac{\tau}{2} \|x - x_g\|^2, \ \forall x \in \mathcal{X}$$

*for some $\tau > 0$. If $\|f - g\|_{\sup} \leq \varepsilon$, then*

$$\|x_f - x_g\| \leq 2\left(\frac{\varepsilon}{\tau}\right)^{1/2}.$$

*Proof.* Note that $g(x_f) - \varepsilon \leq f(x_f) \leq f(x_g) \leq g(x_g) + \varepsilon$, hence

$$g(x_f) - g(x_g) \leq 2\varepsilon.$$

By strong convexity, $2\varepsilon \geq \frac{\tau}{2}\|x_f - x_g\|^2$, which after rearranging gives the desired conclusion. □

The above lemma, along with Propositions 8 and Taylor's theorem immediately yields the following.

**Lemma 11.** *Assume HP transfer holds (Def. 3), then $b_n = O\left(\bar{a}_n^{1/2}\right)$ and $c_n = \Theta(b_n^2)$.*

To relate the loss gap $a_n$ and the uniform loss gap $\bar{a}_n$ we must make further assumptions. Intuitively, we would like to capture the fact that typically the uniform loss gap is dominated by a fairly uniform, positive loss gap for HPs which are nearly optimal.

**Definition 4.** *We will say that the uniform loss gap is locally tight if there exists some radius $\bar{r}$ and constants $0 < c \leq C$ such that for all $\nu \in B(\nu^\star(\infty), \bar{r})$,*

$$c\bar{a}_n \leq \phi_n(\nu) - \phi_\infty(\nu) \leq C\bar{a}_n.$$

This essentially states that the uniform loss gap tightly controls the convergence rate for any nearly optimal set of HPs. This corresponds with the empirical observation that nearly optimal hyperparameters obey identical scaling laws. Under this assumption it is easy to see that $a_n = \Theta(\bar{a}_n)$.

**Lemma 12.** *Assume HP transfer holds (Def. 3). If the uniform loss gap $\bar{a}_n$ is locally tight then $a_n = \Theta(\bar{a}_n)$.*

*Proof.* Note that we have

$$\phi_n(\boldsymbol{\nu}^\star(n)) - \phi_\infty(\boldsymbol{\nu}^\star(\infty)) = \phi_n(\boldsymbol{\nu}^\star(n)) - \phi_\infty(\boldsymbol{\nu}^\star(n)) + \phi_\infty(\boldsymbol{\nu}^\star(n)) - \phi_\infty(\boldsymbol{\nu}^\star(\infty))$$
$$\geq \phi_n(\boldsymbol{\nu}^\star(n)) - \phi_\infty(\boldsymbol{\nu}^\star(n)),$$
$$\phi_n(\boldsymbol{\nu}^\star(n)) - \phi_\infty(\boldsymbol{\nu}^\star(\infty)) = \phi_n(\boldsymbol{\nu}^\star(n)) - \phi_n(\boldsymbol{\nu}^\star(\infty)) + \phi_n(\boldsymbol{\nu}^\star(\infty)) - \phi_\infty(\boldsymbol{\nu}^\star(\infty))$$
$$\leq \phi_n(\boldsymbol{\nu}^\star(\infty)) - \phi_\infty(\boldsymbol{\nu}^\star(\infty)).$$

Since $\boldsymbol{\nu}^\star(n) \to \boldsymbol{\nu}^\star(\infty)$ we can apply the inequalities in Definition 4 for large enough $n$ to yield the claim. $\qquad\square$

*Proof of Proposition 1.* This follows directly by applying Lemmas 11 and 12 $\qquad\square$

## C.3 GRID SEARCH

We now turn towards the connection between the previous asymptotic quantities and compute-optimal grid search. Define a *grid* $\mathcal{G}$ in a search space $\mathcal{X}$ to be a collection of points $\{\boldsymbol{\nu}^{(1)}, \ldots, \boldsymbol{\nu}^{(M)}\}$ contained in $\mathcal{X}$. The *grid resolution* $\rho(\mathcal{G}, \mathcal{X})$ is defined as the largest distance of a point in $\mathcal{X}$ to a point in $\mathcal{G}$, that is

$$\rho(\mathcal{G}, \mathcal{X}) := \sup_{\boldsymbol{\nu} \in \mathcal{X}} \min_{\boldsymbol{\nu}' \in \mathcal{G}} \|\boldsymbol{\nu} - \boldsymbol{\nu}'\|.$$

For a grid $\mathcal{G}$ in the search space $\mathcal{X}$, define $\boldsymbol{\nu}^\star(n, \mathcal{G}) = \arg\min_{\boldsymbol{\nu} \in \mathcal{G}} \phi_n(\boldsymbol{\nu})$. Let us assume that we are allocated a flops budget $\mathcal{F}$ in order to perform hyperparameter search and produce a final model.

Recall that for brevity we use $f(x) \sim g(x)$ to mean $f(x) = \Theta(g(x))$. For a grid $\mathcal{G}$ of resolution $\rho$, we will make the following convenience assumption for Theorem 2 .

**Assumption 1** (Grid proximity). *For a grid $\mathcal{G}$ of resolution $\rho$, we assume that*

$$\min_{\boldsymbol{\nu} \in \mathcal{G}} \|\boldsymbol{\nu} - \boldsymbol{\nu}^\star(n)\| \sim \rho.$$

This assumption is morally true if $\mathcal{G}$ is not chosen with knowledge of the location of $\boldsymbol{\nu}^\star(\infty)$. If for a given $\mathcal{G}$ we chose $\boldsymbol{\nu}^\star(\infty)$ uniformly at random within $\mathcal{G}$, then this assumption holds on average. Suppose we have a compute budget of $\mathcal{F}$ flops, and the amount of flops needed for a single training run scales with $n^r$ for models of width $n$, where $r = 2$ for standard optimization algorithms on standard architectures. For a scaling $\boldsymbol{\gamma}$, we will evaluate the quality of a set of HPs $\boldsymbol{\nu}$ by performing a full training run for a certain width $n$ using the scaled hyperparameters $\mathcal{H}_n(\boldsymbol{\nu}, \boldsymbol{\gamma})$. Let us assume we perform a grid search over $h$ HPs. We first consider compute optimal performance when directly tuning the HPs on a large model.

*Proof of Theorem 2(a).* For a grid of resolution $\rho = \rho(\mathcal{G}, \mathcal{X})$ we will have $|\mathcal{G}| \sim \rho^{-h}$ and $\mathcal{F} \sim n^r \rho^{-h}$. Now observe that by uniform convergence of derivatives (Prop. 8), for $n$ large enough $\phi_n$ will satisfy $(\tau', \delta')$-LSC for some constants $\tau', \delta' > 0$ and so $\phi_n(\boldsymbol{\nu}^\star(n, \mathcal{G})) - \phi_n(\boldsymbol{\nu}^\star(n)) \sim \rho^2$ by Assumption 1. Therefore,

$$\phi_n(\boldsymbol{\nu}^\star(n, \mathcal{G})) - \phi_\infty^\star = \phi_n(\boldsymbol{\nu}^\star(n, \mathcal{G})) - \phi_n(\boldsymbol{\nu}^\star(n)) + \phi_n(\boldsymbol{\nu}^\star(n)) - \phi_\infty^\star$$
$$\sim \rho^2 + n^{-\alpha}$$
$$\sim n^{2r/h}\mathcal{F}^{-2/h} + n^{-\alpha}.$$

We see the final expression is minimized by taking $n^\star \sim \mathcal{F}^{\frac{2}{h\alpha+2r}}$ which yields the rate

$$\phi_n(\boldsymbol{\nu}^\star(n, \mathcal{G})) - \phi_\infty^\star \sim \mathcal{F}^{-\frac{2\alpha}{h\alpha+2r}},$$

as claimed. $\qquad\square$

Now we consider the strategy of transferring the optimal HPs from a smaller model. We say that transfer is *useful* if this strategy achieves a better loss scaling than directly tuning the large model under the same compute budget, as specified above.

*Proof of Theorem 2(b).* Note that in this setting $\mathcal{F} \sim n^r \rho^{-h} + M^r$. The performance scaling is

$$
\begin{aligned}
\phi_M(\boldsymbol{\nu}^\star(n, \mathcal{G})) - \phi_\infty^\star &= \phi_M(\boldsymbol{\nu}^\star(n, \mathcal{G})) - \phi_M(\boldsymbol{\nu}^\star(n)) \\
&\quad + \phi_M(\boldsymbol{\nu}^\star(n)) - \phi_M(\boldsymbol{\nu}^\star(M)) \\
&\quad + \phi_M(\boldsymbol{\nu}^\star(M)) - \phi_\infty^\star \\
&\sim \rho^2 + n^{-2\beta} + M^{-\alpha} \\
&\sim \left(\frac{n^r}{\mathcal{F} - M^r}\right)^{2/h} + n^{-2\beta} + M^{-\alpha}.
\end{aligned}
$$

Since $\mathcal{F} - M^r \sim \mathcal{F}$, we should take $M^\star \sim \mathcal{F}^{1/r}$ in which case the above simplifies to

$$
\phi_M(\boldsymbol{\nu}^\star(n, \mathcal{G})) - \phi_\infty^\star \sim \frac{n^{2r/h}}{\mathcal{F}^{2/h}} + n^{-2\beta} + \mathcal{F}^{-\alpha/r}
$$

which is minimized a $n^\star \sim \mathcal{F}^{\frac{1}{h\beta+r}}$ and yields $\phi_M(\boldsymbol{\nu}^\star(n, \mathcal{G})) - \phi_\infty^\star \sim \mathcal{F}^{\frac{-2\beta}{h\beta+r}} + \mathcal{F}^{\frac{-\alpha}{r}}$. Now note that $\frac{\alpha}{r} > \frac{2\alpha}{h\alpha+2r}$ and $\frac{2\beta}{h\beta+r} > \frac{2\alpha}{h\alpha+2r}$ if and only if $\beta > \alpha/2$, which is the condition for useful transfer. $\qquad\square$

## C.4 RANDOM FEATURES REGRESSION

Consider the following data generating process where the labels come from a single-index model, and we train a random features ridge regression estimator on $N$ samples,

$$
\begin{aligned}
y &= \sigma_*(\langle \boldsymbol{x}, \boldsymbol{\beta}_* \rangle) + \varepsilon, \quad \text{where } \boldsymbol{x} \sim \mathcal{N}(0, \boldsymbol{I}_d), \|\boldsymbol{\beta}_*\| = 1, \text{Var}(\varepsilon) = \sigma_\varepsilon^2. \\
f(\boldsymbol{x}) &= \langle \boldsymbol{a}_\lambda, \sigma(\boldsymbol{W}\boldsymbol{x}) \rangle, \quad \text{where } \boldsymbol{W} \in \mathbb{R}^{d \times n}, [\boldsymbol{W}]_{i,j} \sim \mathcal{N}(0, 1/d), \\
\boldsymbol{a}_\lambda &:= \text{argmin}_{\boldsymbol{a} \in \mathbb{R}^n} \sum_{i=1}^N (y_i - \langle \boldsymbol{a}, \sigma(\boldsymbol{W}\boldsymbol{x}_i) \rangle)^2 + \lambda \|\boldsymbol{a}\|_2^2.
\end{aligned}
$$

We aim to select the optimal regularization parameter $\lambda$ that minimizes the prediction risk (generalization error) $\mathcal{R} = \mathbb{E}_{\boldsymbol{x}}[(y - f(\boldsymbol{x}))^2]$. We make the following assumptions.

**Assumption 2.**

- *Proportional limit.* $N, d, n \to \infty$, $N/d \to \psi_1$, $n/d \to \psi_2$ where $\psi_1, \psi_2 \in (0, \infty)$.

- *Normalized activation.* *Both the student and teacher nonlinearities are normalized such that* $\mathbb{E}[\sigma], \mathbb{E}[\sigma_*] = 0$, $\|\sigma\|_\gamma, \|\sigma_*\|_\gamma = 1$, *and also* $\|\sigma'\|_\gamma, \|\sigma_*'\|_\gamma \neq 0$. *We further require that $\sigma$ is a nonlinear odd function with bounded first three derivatives, and $\sigma_*$ is $\Theta(1)$-Lipschitz.*

**Remark 1.** *The above assumptions are standard in the high-dimensional asymptotic analysis of random features models, see e.g., [MM22; GLK+20]. The non-zero expectation of $\sigma', \sigma_*'$ is necessary for the RF model to outperform the null estimator in the proportional regime. The assumption of odd $\sigma$ simplifies the Gaussian equivalence computation — see [HL22; BES+22].*

**Asymptotic prediction risk.** Under Assumption 2, following [HL22], we know that the asymptotic prediction risk is given as by the following implicit equations,

$$
\lim_{n,d,N \to \infty} \mathbb{E}[(y - f(\boldsymbol{x}))^2] \xrightarrow{\mathbb{P}} \mathcal{R}(\lambda) := -(\mu_2^{*2} + \sigma_\varepsilon^2) \cdot \frac{m_1'(\lambda)}{m_1(\lambda)^2} - \mu_1^{*2} \cdot \frac{m_2'(\lambda)}{m_1(\lambda)^2},
$$

where the Hermite coefficients $\mu_1^* = \mathbb{E}_{z \sim \mathcal{N}(0,1)}[\sigma_*'(z)]$, $\mu_2^{*2} = 1 - \mu_1^{*2}$, and the coupled Stieltjes transforms $m_1(z)$ and $m_2(z) \in \mathbb{C}^+ \cup \mathbb{R}_+$ are uniquely defined by the following self-consistent equations for $z \in \mathbb{C}^+ \cup \mathbb{R}_+$,

$$
\frac{1}{\psi_1}(m_1(z) - m_2(z))(\mu_2^2 m_1(z) + \mu_1^2 m_2(z)) + \mu_1^2 m_1(z) m_2(z)(z m_1(z) - 1) = 0, \tag{9}
$$

$$
\frac{\psi_2}{\psi_1}\left(\mu_1^2 m_1(z) m_2(z) + \frac{1}{\psi_1}(m_2(z) - m_1(z))\right) + \mu_1^2 m_1(z) m_2(z)(z m_1(z) - 1) = 0, \tag{10}
$$

where $\mu_1 = \mathbb{E}_{z \sim \mathcal{N}(0,1)}[\sigma'(z)]$, $\mu_2^2 = 1 - \mu_1^2$. Note that $\sigma$ being nonlinear implies $\mu_2 \neq 0$. We omit the argument in $m_1(\lambda), m_2(\lambda)$ except when tracking the $\lambda$-dependence. $m_1', m_2'$ stand for derivative with respect to $\lambda$. To further simplify the exposition, we define $\eta := \psi_1/\psi_2$, and write the asymptotic prediction risk at width $\psi_2$ and ridge penalty $\lambda$ as $\mathcal{R}_{\psi_2}(\lambda) = \mathcal{R}_{\psi_1/\eta}(\lambda)$.

**Large-width limit.** First we consider the test performance of the "infinite-width" model, which corresponds to taking $\psi_2 = n/d \to \infty$ or $\eta \to 0$. Note that the prediction risk in this limit is well-defined and has been computed in prior works (see e.g., [BMR21]). First recall that $m_1, m_2 > 0$ and $zm_1(z) - 1$ remains uniformly bounded for any $1/\eta$, hence from (10) we know that at the large-width limit,

$$\mu_1^2 m_1 m_2 + \psi_1^{-1}(m_2 - m_1) =: \mathscr{T}_1(m_1, m_2) = 0, \quad \lambda m_1 + \mu_2^2 m_1 + \mu_1^2 m_2 - 1 =: \mathscr{T}_2(m_1, m_2, \lambda) = 0.$$

Reparameterize $t := 1 + \mu_1^2 \psi_1 m_1 > 1$, we have $\lambda(t) = \frac{\mu_1^2 \psi_1}{t-1} - \frac{\mu_1^2}{t} - \mu_2^2$. By the chain rule,

$$\partial_t m_1 = \frac{1}{\mu_1^2 \psi_1}, \quad \partial_t m_2 = \frac{1}{\mu_1^2 \psi_1 t^2}, \quad \partial_t \lambda = -\frac{\mu_1^2 S(t)}{t^2(t-1)^2},$$

where we defined $S(t) := \psi_1 t^2 - (t-1)^2$. Hence at $\eta = 0$ we have

$$\frac{m_1'}{m_1^2} = -\frac{\psi_1 t^2}{S(t)}, \quad \frac{m_2'}{m_1^2} = -\frac{\psi_1}{S(t)}.$$

Therefore,

$$\mathcal{R}_\infty(\lambda(t)) := \lim_{\psi_2 \to \infty} \mathcal{R}_{\psi_2}(\lambda(t)) = \frac{\psi_1((\mu_2^{*2} + \sigma_\varepsilon^2)t^2 + \mu_1^{*2})}{S(t)}.$$

Differentiating the risk yields the closed-form expression of the optimal ridge penalty (consistent with [DW18; WX20]),

$$\lambda^*(\infty) = \frac{\mu_1^2(\sigma_\varepsilon^2 + \mu_2^{*2})}{\mu_1^{*2}} - \mu_2^2. \tag{11}$$

We restrict ourself to the setting where non-vanishing regularization is needed at the large-width limit, i.e., $\lambda^*(\infty) > 0$. Denote $t_*$ as the corresponding optimal value of $t = 1 + \mu_1^2 \psi_1 m_1$, the optimality condition $\mu_1^{*2} = \frac{(\mu_2^{*2} + \sigma_\varepsilon^2)t_*(t_* - 1)}{\psi_1 t_* - t_* + 1}$ implies that

$$\psi_1 t_* - t_* + 1 > 0, \quad S(t_*) = (\psi_1 t_* - t_* + 1)t_* + (t_* - 1) > 0.$$

Hence we have the following characterization of the curvature

$$\mathcal{R}_\infty''(\lambda^*(\infty)) = \frac{2(\mu_2^{*2} + \sigma_\varepsilon^2)\psi_1}{S(t_*)\lambda'(t_*)^2(\psi_1 t_* - t_* + 1)} > 0, \quad \lambda'(t_*) = -\frac{\mu_1^2 S(t_*)}{t_*^2(t_* - 1)^2} < 0. \tag{12}$$

Note that (12) validates the local strong convexity of $\mathcal{R}_\infty$.

**Finite-width sensitivity.** Now consider the system given by (9)(10)

$$E(m_1, m_2, \lambda, \eta) := \begin{bmatrix} \mathscr{T}_2(m_1, m_2, \lambda) \\ \mathscr{T}_1(m_1, m_2) + \eta \mathscr{T}_3(m_1, m_2, \lambda) \end{bmatrix},$$

where $\mathscr{T}_3 = \mu_1^2 m_1 m_2(\lambda m_1 - 1)$. Differentiate $E = 0$ with respect to $\eta$ and evaluate at $\eta = 0$,

$$J_0 \begin{bmatrix} \partial_\eta m_1 \\ \partial_\eta m_2 \end{bmatrix} = -\begin{bmatrix} 0 \\ T \end{bmatrix}\bigg|_{\eta=0}, \quad \text{where} \quad J_0 := \partial_{m_1, m_2}(\mathscr{T}_2, \mathscr{T}_1) = \begin{bmatrix} \lambda + \mu_2^2 & \mu_1^2 \\ \mu_1^2 m_2 - \psi_1^{-1} & \mu_1^2 m_1 + \psi_1^{-1} \end{bmatrix}. \tag{13}$$

Recall that $\mathscr{T}_2 = 0$ yields $\lambda m_1 - 1 = -(\mu_2^2 m_1 + \mu_1^2 m_2)$, and hence $T|_{\eta=0} = -\mu_1^2 m_1 m_2(\mu_2^2 m_1 + \mu_1^2 m_2)$. On the other hand, by direct computation

$$\det J_0 = \frac{\mu_1^2 S(t)}{\psi_1 t(t-1)}, \quad \Rightarrow \quad \det J_0(t_*) > 0. \tag{14}$$

Solving the linear system (13) yields

$$\partial_\eta m_1 = -\frac{\mu_1^4 m_1 m_2(\mu_2^2 m_1 + \mu_1^2 m_2)}{\det J_0} = -\frac{(t-1)^4(\mu_1^2 + \mu_2^2 t)}{\mu_1^4 \psi_1^2 t S(t)},$$

$$\partial_\eta m_2 = -\frac{(\lambda + \mu_2^2)\mu_1^2 m_1 m_2 (\mu_2^2 m_1 + \mu_1^2 m_2)}{\det J_0} = \frac{(t-1)^3(\mu_1^2 + \mu_2^2 t)(\psi_1 t - t + 1)}{\mu_1^4 \psi_1^2 t^2 S(t)}.$$

Differentiating the prediction risk with respect to $\eta$ and evaluate at the large-width limit $\eta = 0$,

$$\partial_{\eta=0}\mathcal{R}_\infty(\lambda) = -(\mu_2^{*2} + \sigma_\varepsilon^2)\left(\frac{\partial_\eta m_1'}{m_1^2} - \frac{2m_1'\partial_\eta m_1}{m_1^3}\right) - \mu_1^{*2}\left(\frac{\partial_\eta m_2'}{m_1^2} - \frac{2m_2'\partial_\eta m_1}{m_1^3}\right),$$

where $\partial_{\eta=0}\mathcal{R}_\infty(\lambda) = \partial_\eta \mathcal{R}_{\psi_1/\eta}(\lambda)\big|_{\eta=0}$. A similar determinant calculation yields

$$\partial_\eta m_1' = -\frac{\mu_1^2 \mathscr{S}}{\det J_0}, \quad \partial_\eta m_2' = -\frac{(\lambda + \mu_2^2)\mathscr{S}}{\det J_0},$$

where

$$\mathscr{S} := \mu_1^2 m_2 (2\lambda m_1 - 1) m_1' + \mu_1^2 m_1 (\lambda m_1 - 1) m_2' + \mu_1^2 m_1^2 m_2$$
$$= \frac{(t-1)^3}{\mu_1^4 \psi_1^2}\left[-\frac{t}{S(t)} + \frac{(2t+1)}{S(t)} \cdot \frac{t-1}{\mu_1^2 \psi_1} \cdot \frac{\mu_1^2 + \mu_2^2 t}{t}\right] + \frac{(t-1)^3}{\mu_1^4 \psi_1^3 t}.$$

Using the above, a tedious algebraic calculation gives the following expression of the sensitivity of the prediction risk (at fixed $\lambda$) with respect to $\eta$:

$$\partial_{\eta=0}\mathcal{R}_\infty(\lambda) = \frac{(t-1)^2 Q(t)}{\mu_1^2 S(t)^2}, \tag{15}$$

where $Q(t) = \sum_{k=0}^2 c_k t^k$ with coefficients

$$c_2 = \mu_1^2(\mu_2^{*2} + \sigma_\varepsilon^2) - \mu_2^2(\mu_2^{*2} + \sigma_\varepsilon^2) + 2\mu_2^2\mu_1^{*2}(\psi_1 - 1),$$
$$c_1 = -3\mu_1^2(\mu_2^{*2} + \sigma_\varepsilon^2) + \mu_1^2\mu_1^{*2}(\psi_1 - 1) + \mu_2^2(\mu_2^{*2} + \sigma_\varepsilon^2) + \mu_2^2\mu_1^{*2}(\psi_1 + 3),$$
$$c_0 = 2\mu_1^2(\mu_2^{*2} + \sigma_\varepsilon^2) + \mu_1^{*2}(2\mu_1^2\psi_1 + 2\mu_1^2 - 1).$$

Importantly, under the optimal $\lambda$ defined in (11), we have the factorization

$$Q(t_*) = \frac{2(\mu_2^{*2} + \sigma_\varepsilon^2)(\mu_1^2 + \mu_2^2 t_*)(t_* - 1)S(t_*)}{\psi_1 t_* - t_* + 1},$$

and since $(t_* - 1), S(t_*), (\psi_1 t_* - t_* + 1) > 0$, we conclude the derivative (15) at $t_*$ is strictly positive

$$\partial_{\eta=0}\mathcal{R}_\infty(\lambda_*(\infty)) = \frac{2(\mu_2^{*2} + \sigma_\varepsilon^2)(\mu_1^2 + \mu_2^2 t_*)(t_* - 1)^3}{\mu_1^2 S(t_*)(\psi_1 t_* - t_* + 1)} =: C_\eta > 0. \tag{16}$$

**Putting things together.** Recall that the asymptotic prediction risk $\mathcal{R}$ is $C^2$ in $\lambda$ and $C^1$ in $\eta$. Given the Jacobian invertibility (14) and local strong convexity (12), the implicit function theorem (IFT) implies that there exists a neighborhood defined by some $\eta_0 > 0$ and a unique $C^1$ map $\bar\lambda^* : [0, \eta_0) \to \mathbb{R}_+$, such that $\bar\lambda^*(0) = \lambda^*(\infty)$, $\partial_\lambda \mathcal{R}_\eta(\bar\lambda^*(\eta)) = 0$, and $\partial_\lambda^2 \mathcal{R}_\eta(\bar\lambda^*(\eta)) > 0$. Consequently, we may take a first-order expansion and conclude (setting $\eta = \psi_1/\psi_2$ under with $\psi_1$)

$$\lambda^*(\psi_2) = \lambda^*(\infty) + \partial_{\eta \to 0_+}\bar\lambda^*(\eta) \cdot \frac{\psi_1}{\psi_2} + o(\psi_2^{-1}), \quad \partial_{\eta \to 0_+}\bar\lambda^*(\eta) := -\frac{\partial_\lambda \partial_\eta \mathcal{R}_\infty(\lambda^*(\infty))}{\partial_\lambda^2 \mathcal{R}_\infty(\lambda^*(\infty))} =: C_\lambda. \tag{17}$$

Note that the denominator in $C_\lambda$ is strictly positive by (12). Moreover, since $\lambda'(t_*) \neq 0$, we may write $\partial_\lambda \partial_\eta \mathcal{R}_\infty(\lambda) = \frac{\partial_t(\partial_{\eta=0}\mathcal{R}(\lambda(t)))}{\partial_t \lambda(t)}$, and compute $C_\lambda = \frac{(3\psi_1 - 4\mu_1^2\psi_1 + 1)t_*^2 + 2(2\mu_1^2\psi_1 - 1)t_* + 1}{2\psi_1 t_*^2}$. This confirms that the hyperparameter gap vanishes at a rate of $O(\psi_2^{-1})$.

For the loss gap, denote $\delta_\eta = \lambda^*(\psi_1/\eta) - \lambda^*(\infty)$ for $\eta \in [0, \eta_0)$, the IFT and Taylor expansion gives

$$\mathcal{R}_{\psi_1/\eta}(\lambda^*(\infty) + \delta_\eta) = \mathcal{R}_\infty(\lambda^*(\infty)) + \partial_{\lambda=\lambda^*(\infty)}\mathcal{R}_\infty(\lambda^*(\infty))\delta_\eta + \partial_{\eta=0}\mathcal{R}_\infty(\lambda^*(\infty)) + O(\delta_\eta^2 + \eta^2)$$

$$\overset{(i)}{=} \mathcal{R}_\infty(\lambda^*(\infty)) + C_\eta \cdot \frac{\psi_1}{\psi_2} + o(\psi_2^{-1}),$$

where $(i)$ is due to the stationarity condition $\partial_{\lambda=\lambda^*(\infty)}\mathcal{R}_\infty(\lambda^*(\infty)) = 0$ and $\delta_\eta = O(\eta)$ from (17), and $C_\eta > 0$ is explicitly given in (16). The strict positivity of $C_\eta$ ensures that the loss gap scales exactly as $\Theta(\psi_2^{-1})$. Hence by Proposition 1 and Theorem 2 we know that the ridge penalty in RF regression exhibits *fast and useful transfer*, i.e., the suboptimality gap $|\mathcal{R}_\infty(\lambda^*(\psi_2)) - \mathcal{R}_\infty(\lambda^*(\infty))| \sim \psi_2^{-2} \ll |\mathcal{R}_{\psi_2}(\lambda^*(\psi_2)) - \mathcal{R}_\infty(\lambda^*(\infty))|$, which aligns with the observations in Figure 4 and concludes Theorem 3.

## C.5 Decomposition-aware Fast Transfer

In this section we formalize our quantitative bound on the HP gap $b_n$ in terms of the top-$\kappa$ invariance and residual flatness arising from our decomposition. For the sake of simplicity we will assume that $\mathcal{X}$ is small enough so that the local strong convexity condition holds globally.

**Definition 5.** *For $f \in C^2(\mathcal{X})$, define the curvature $\mu(f)$ and the Lipschitz constant $\mathrm{Lip}(f)$ as:*

$$\mu(f) := \inf_{\boldsymbol{\nu} \in \mathcal{X}} f''(\boldsymbol{\nu}), \ \ \mathrm{Lip}(f) := \sup_{\boldsymbol{\nu} \in \mathcal{X}} |f'(\boldsymbol{\nu})|.$$

**Definition 6** (Decomposition Rate)**.** *Let $\phi_n$ and $\mathcal{X}$ be as in Definition 3, and $\phi_n^\kappa$ and $\boldsymbol{\nu}_\kappa^\star(n)$ as in Eq. (5). We define the following quantities associated with the decomposition:*

- ***Top-$\kappa$ invariance gap:*** $\varepsilon_{\mathrm{inv}}(n, \kappa) := \frac{\|\phi_n^\kappa - \phi_\infty^\kappa\|_{\sup}}{\mu(\phi_n^\kappa) \vee \mu(\phi_\infty^\kappa)}$

- ***Residual flatness gap:*** $\varepsilon_{\mathrm{flat}}(n, \kappa) := \frac{\mathrm{Lip}(\phi_n^{-\kappa})}{\mu(\phi_n)} + \frac{\mathrm{Lip}(\phi_\infty^{-\kappa})}{\mu(\phi_\infty)}$

- ***Decomposition objective:*** $\mathcal{J}(\kappa) := 2\sqrt{\varepsilon_{\mathrm{inv}}(n, \kappa)} + \varepsilon_{\mathrm{flat}}(n, \kappa)$

- ***Decomposition HP gap:*** $t_n := \min_\kappa \mathcal{J}(\kappa)$ s.t. $\mu(\phi_n^\kappa) \wedge \mu(\phi_\infty^\kappa) > 0$.

The decomposition HP gap $t_n$ is defined to be a natural upper bound on the HP gap $b_n$ as we show in Proposition 13 which makes use of Lemmas 10 and 14. This upper bound is obtained by choosing an optimal HP dependent truncation index $\kappa_n^\star$ minimizing $\mathcal{J}(\kappa)$ such that $\varepsilon_{\mathrm{inv}}(n, \kappa_n^\star)$ which quantifies top-$\kappa$ invariance and $\varepsilon_{\mathrm{flat}}(n, \kappa_n^\star)$ which quantifies residual flatness are both appropriately small. Note that $t_n$ is well-defined for $n$ large enough because we can take $\kappa \equiv n$ and from the assumptions of Definition 3 both $\phi_n$ and $\phi_\infty$ are strongly convex.

**Proposition 13.** *Assume the setting of Definition 3 where the local strong convexity is global. The decomposition HP gap $t_n$ in Definition 6 satisfies $t_n \geq b_n$ where $b_n$ is the HP gap from Definition 1.*

We remark that we introduce the quantity $t_n$ primarily as a theoretical quantity for conceptual purposes. The quantity $t_n$ will be small when top-$\kappa$ invariance and residual flatness holds and since $t_n \geq b_n$ this will imply $b_n$ is small as well. We also note that it is natural to chose the optimal truncation index $\kappa_n^\star$ used in $t_n$ to be a function of the width $n$. This is because as $n \to \infty$ we expect $b_n \to 0$ and so it is desirable that $t_n \to 0$ as well which will not be the case if we used a fixed $\kappa$ because $\varepsilon_{\mathrm{flat}}$ in $\mathcal{J}(\kappa)$ will converge to a non-zero value. Overall, one can view $t_n$ being small[3] as an explicit sufficient condition for fast transfer. We conjecture that (some version of) this condition holds in practice when training neural networks on natural data with optimizers like Adam or SGD. For future work, it would be interesting to provide natural settings where such a formal statement is provably true.

To prove Proposition 13 we will first need a perturbation result similar to Lemma 10.

**Lemma 14.** *Let $f : \mathcal{X} \to \mathbb{R}$ and $g : \mathcal{X} \to \mathbb{R}$ such that $f$ is $\tau$ strongly-convex and $\sup_{\boldsymbol{x} \in \mathcal{X}} |g'(\boldsymbol{x})| \leq \varepsilon$. Let $\boldsymbol{x}^\star = \arg \min_{\boldsymbol{x} \in \mathcal{X}} f(\boldsymbol{x})$ and $\tilde{\boldsymbol{x}} = \arg \min_{\boldsymbol{x} \in \mathcal{X}} f(\boldsymbol{x}) + g(\boldsymbol{x})$. Then*

$$\|\boldsymbol{x}^\star - \tilde{\boldsymbol{x}}\| \leq \frac{\varepsilon}{\tau}.$$

*Proof.* By first order optimality we have $f'(\tilde{\boldsymbol{x}}) + g'(\tilde{\boldsymbol{x}})$, hence $|f'(\tilde{\boldsymbol{x}})| = |g'(\tilde{\boldsymbol{x}})| \leq \varepsilon$. By strong convexity $\tau\|\tilde{\boldsymbol{x}} - \boldsymbol{x}^\star\| \leq |f'(\tilde{\boldsymbol{x}})| \leq \varepsilon$ which gives the result. $\square$

*Proof of Proposition 13.* For a given $n$ and $\kappa$ such that $\mu(\phi_n^\kappa) \wedge \mu(\phi_\infty^\kappa) > 0$,

$$\begin{aligned}
b_n &= \|\boldsymbol{\nu}^\star(n) - \boldsymbol{\nu}^\star(\infty)\| \\
&= \|\boldsymbol{\nu}^\star(n) - \boldsymbol{\nu}_\kappa^\star(n) + \boldsymbol{\nu}_\kappa^\star(n) - \boldsymbol{\nu}_\kappa^\star(\infty) + \boldsymbol{\nu}_\kappa^\star(\infty) - \boldsymbol{\nu}^\star(\infty)\| \\
&\leq \|\boldsymbol{\nu}^\star(n) - \boldsymbol{\nu}_\kappa^\star(n)\| + \|\boldsymbol{\nu}_\kappa^\star(n) - \boldsymbol{\nu}_\kappa^\star(\infty)\| + \|\boldsymbol{\nu}_\kappa^\star(\infty) - \boldsymbol{\nu}^\star(\infty)\| \\
&\leq 2\sqrt{\varepsilon_{\mathrm{inv}}(n, k)} + \|\boldsymbol{\nu}^\star(n) - \boldsymbol{\nu}_\kappa^\star(n)\| + \|\boldsymbol{\nu}_\kappa^\star(\infty) - \boldsymbol{\nu}^\star(\infty)\| \\
&\leq 2\sqrt{\varepsilon_{\mathrm{inv}}(n, \kappa)} + \varepsilon_{\mathrm{flat}}(n, \kappa),
\end{aligned}$$

---

[3] Relative to the bound on $b_n$ implied solely by $a_n$ (see Proposition 1).

where the first inequality is the triangle inequality, the second comes from Lemma 10, and the last inequality comes from Lemma 14. Taking the minimum of the right hand side over valid $\kappa$ yields the claim $t_n \geq b_n$. $\qquad\square$

# D  TRUNCATION INDEX SELECTION

Empirically finding the minimizer $\kappa^\star(n)$ in the definition of $t_n$ (Def. 6) is not tractable due to complicated nature of the decomposition objective $\mathcal{J}$. Instead of trying to minimize $\mathcal{J}$, we will use a simpler surrogate process which we outline below.

Note that given a finite grid of HPs $\{\nu_1, \ldots, \nu_g\}$, we only need to produce a truncation index for each $\nu_i$ with $i \in [g]$. Let $\boldsymbol{\kappa} = (\kappa_1, \ldots, \kappa_g)$ represent the vector of these values where $\kappa_i = \kappa(\nu_i)$ for $i \in [g]$. We define $\phi_n^{\boldsymbol{\kappa}}$ to be the set of pointwise evaluations $\{(\nu_i, \phi_n^{\kappa_i}(\nu_i))\}_{i \in [g]}$ and identify it with the function obtained from its linear interpolation. Our goal is to return a set of truncation index vectors $\hat{\boldsymbol{\kappa}}(n)$.

Let $n_{\max}$ denote the largest width model under consideration and fix a width $n < n_{\max}$. Consider the following proxy objective with parameters $\boldsymbol{\tau} = (\tau_1, \tau_2)$ and $\tau_1, \tau_2 > 0$,

$$\mathcal{J}_{\text{proxy}}(\boldsymbol{\kappa}; \boldsymbol{\tau}) := \frac{1}{g} \sum_{i \in [g]} |\phi_n^{\kappa_i}(\nu_i) - \phi_{n_{\max}}^{\kappa_i}(\nu_i)| + \tau_1 \cdot \text{Lip}(\phi_n^{-\boldsymbol{\kappa}}) + \tau_2 \cdot \text{Lip}(\phi_{n_{\max}}^{-\boldsymbol{\kappa}}), \quad (18)$$

and define its minimizer to be $\boldsymbol{\kappa}^\star(\boldsymbol{\tau}) := \arg\min_{\boldsymbol{\kappa}} \mathcal{J}_{\text{proxy}}(\boldsymbol{\kappa}; \boldsymbol{\tau})$, which can be found approximately using coordinate descent (see Algorithm 2). The objective $\mathcal{J}_{\text{proxy}}$ is similar to the objective $\mathcal{J}$ in Definition 6, except that instead of a supnorm we use an average $\ell_1$-norm to promote tractability, we use $n_{\max}$ as an infinite-width proxy, and we absorb all the curvature based scalings $\mu(\cdot)$ into constants $\tau_1, \tau_2$. We will set $\hat{\kappa}(n) = \boldsymbol{\kappa}^\star(\hat{\boldsymbol{\tau}})$ for a "reasonable" choice of $\hat{\boldsymbol{\tau}}$. In particular, $\hat{\boldsymbol{\tau}}$ will be chosen to be the smallest[4] $\boldsymbol{\tau}$ so that $\phi_n^{\boldsymbol{\kappa}^\star(\boldsymbol{\tau})}$ and $\phi_{n_{\max}}^{\boldsymbol{\kappa}^\star(\boldsymbol{\tau})}$ are approximately convex and the minimizers are close to the minimizers of $\phi_n$ and $\phi_{n_{\max}}$ respectively, assuming such $\hat{\boldsymbol{\tau}}$ exists. The full details of the process are given in Algorithm 1 in Appendix D.

In this section we describe our procedure (Algorithm 1) for selecting $\hat{\boldsymbol{\kappa}}(n)$ (see Section 4.1). We do not claim this procedure is optimal in any sense and emphasize that we are just searching for a valid $\hat{\boldsymbol{\kappa}}(n)$ so that a certain sufficient condition holds qualitatively in order to support our conjecture for fast hyperparameter transfer. For convenience, we only consider the case where we sweep a single HP, although this can be straightforwardly extended.

As part of our algorithm, we need to measure the convexity and flatness of a function $f$ given a set of pointwise evaluations $\{(\nu_i, y_i)\}_{i=1}^g$ where $\nu_1 < \cdots < \nu_g$ and $y_i = f(\nu_i)$. For each interior index $i = 2, \ldots, g-1$, define the three-point second-derivative estimate

$$\hat{f}''(\nu_i) := 2\left(\frac{y_{i-1}}{(\nu_{i-1} - \nu_i)(\nu_{i-1} - \nu_{i+1})} + \frac{y_i}{(\nu_i - \nu_{i-1})(\nu_i - \nu_{i+1})} + \frac{y_{i+1}}{(\nu_{i+1} - \nu_{i-1})(\nu_{i+1} - \nu_i)}\right).$$

The *convexity error* is the fraction of interior points with negative curvature estimate:

$$\text{ConvErr}(\{(\nu_i, y_i)\}_{i=1}^g) := \frac{1}{g-2} \sum_{i=2}^{g-1} \mathbf{1}\left\{\hat{f}''(\nu_i) < 0\right\}. \quad (19)$$

The *Lipschitz constant* is the maximum slope magnitude:

$$\text{Lip}(\{(\nu_i, y_i)\}_{i=1}^g) := \max_{1 \leq i \leq g-1} \left|\frac{y_{i+1} - y_i}{\nu_{i+1} - \nu_i}\right|. \quad (20)$$

---

[4]For definiteness, we order $(\tau_1, \tau_2)$ by their sum, breaking ties in the first-coordinate.

---

**Algorithm 1:** Compute $\hat{\boldsymbol{\kappa}}(n)$

---

**Input:**

| | |
|---|---|
| $\mathcal{N}$ | set of widths, with $n_{\max} = \max \mathcal{N}$ |
| $\{\nu_i\}_{i=1}^g$ | grid of $g$ hyperparameter points |
| $\mathcal{T}$ | candidate list of $\tau$ values |
| $\varepsilon_{\text{cvx}}, \varepsilon_{\text{amin}}$ | tolerances (convexity, argmin proximity) |
| $\Phi_n \in \mathbb{R}^{g \times n}$ | arrays $\phi_n^k(\nu_i)$ for each $n \in \mathcal{N}$ |

**Output:** $\hat{\boldsymbol{\kappa}}(n)$ truncation vectors in $[n]^g$ for $n < n_{\max}$, or FAIL

1. **Grid:** $\mathcal{G}_{\boldsymbol{\tau}} := \{(\tau_1, \tau_2) : \tau_1, \tau_2 \in \mathcal{T}\}$ sorted by $\tau_1 + \tau_2$

2. **For each** $n \in \mathcal{N} \setminus \{n_{\max}\}$**:**

   (a) **For** $(\tau_1, \tau_2) \in \mathcal{G}_{\boldsymbol{\tau}}$ (in order):

      i. $\boldsymbol{\kappa} \leftarrow$ MINIMIZEPROXY$(\Phi_n, \Phi_{n_{\max}}; \tau_1, \tau_2)$      // Alg. 2; minimizes Eq. (18)

      ii. $E_{\text{cvx}} := \max\{\text{ConvErr}(\phi_n^{\boldsymbol{\kappa}}), \text{ConvErr}(\phi_{n_{\max}}^{\boldsymbol{\kappa}})\}$      // Eq. (19)

      iii. $a_n := \arg\min_\nu \phi_n^\kappa, \quad a_n^{\text{tot}} := \arg\min_\nu \phi_n$;

      $a_\infty := \arg\min_\nu \phi_{n_{\max}}^\kappa, \quad a_\infty^{\text{tot}} := \arg\min_\nu \phi_{n_{\max}}$.

      $\Delta := \max\{|a_n - a_n^{\text{tot}}|, |a_\infty - a_\infty^{\text{tot}}|\}$      // argmin proximity

      iv. **If** $E_{\text{cvx}} \leq \varepsilon_{\text{cvx}}$ **and** $\Delta \leq \varepsilon_{\text{amin}}$, **set** $\hat{\boldsymbol{\kappa}}(n) \leftarrow \boldsymbol{\kappa}$ and **break** to the next $n$.

   (b) **If** no pair in $\mathcal{G}_{\boldsymbol{\tau}}$ is accepted, **set** $\hat{\boldsymbol{\kappa}}(n) \leftarrow$ FAIL.

---

**Algorithm 2:** Minimize Proxy Objective Eq. (18)

---

**Input:**

| | |
|---|---|
| $\Phi_n, \Phi_{n_{\max}}$ | arrays $\phi_n^k(\nu_i)$ and $\phi_{n_{\max}}^k(\nu_i)$, shapes $g \times n$ and $g \times n_{\max}$ |
| $(\tau_1, \tau_2)$ | positive penalty weights |

**Output:** $\boldsymbol{\kappa}^\star \in [n]^g$ (approximate minimizer via coordinate descent)

1. **Initialize:** $\boldsymbol{\kappa} \leftarrow (n/2, \ldots, n/2)$

2. **Repeat** until no coordinate changes:

   **For** $i = 1, \ldots, g$:

      i. *Local scores for each $k \in [n]$:*

      $$\text{score}(k) = \mathcal{J}_{\text{proxy}}(\tilde{\boldsymbol{\kappa}}) \text{ where } \tilde{\boldsymbol{\kappa}} \text{ is } \boldsymbol{\kappa} \text{ with } \kappa_i \text{ switched to } k$$

      ii. *Coordinate update:* $\kappa_i \leftarrow \arg\min_{k \in [n]} \text{score}(k)$.

3. **Return** $\boldsymbol{\kappa}^\star := (\kappa_i)_{i=1}^g$.

---

# E  EXPERIMENTAL DETAILS AND ADDITIONAL EXPERIMENTS

## E.1  $\mu$P MLP EXPERIMENTS

**Experiment Setting.** We consider regression with shallow ReLU network: $f(\boldsymbol{x}) = \sum_{i=1}^{n} a_i \sigma(\langle \boldsymbol{w}_i, \boldsymbol{x} \rangle + b_i)$, where we aim to tune the learning rate $\eta$ that minimizes the validation squared loss. We take the target function to be the following multi-index model on isotropic Gaussian data,

$$ y = \sqrt{\tfrac{d}{4k} \sum_{i=1}^{k} x_i^2}, \quad \boldsymbol{x} \sim \mathcal{N}(0, 4d^{-1}\boldsymbol{I}_d). $$

We set $k = 15$, and $d = 64, n = 2^{15}$. We run the Adam optimizer for $T = 2^{14}$ steps with batch size 256 to minimize the squared loss. The initialization and learning rate are set according to $\mu$P. As shown in Figure 2, the optimal learning rate steadily approaches $\sim 10^{-1}$ before an abrupt jump at width $n = 8192$.

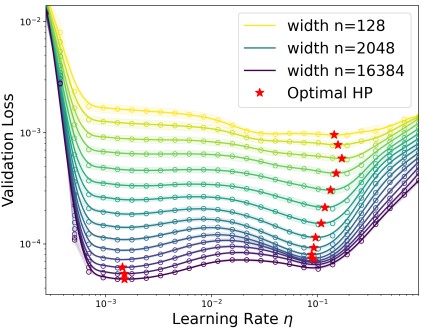 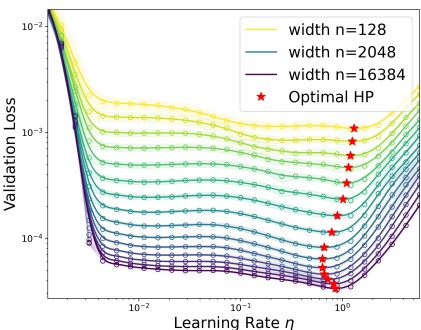

(a) Identical all-layer learning rate (Figure 2).  (b) Different layer-wise learning rate.

Figure 17: Learning Gaussian $k$-index model with two-layer ReLU network under $\mu$P. The HP of interest is the Adam learning rate. **Left:** default $\mu$P implementation with the same learning rate for the first- and second-layer parameters. **Right:** layer-wise learning rate where the second-layer learning rate is divided by 50. Observe that the original $\mu$P yields a "bimodal" HP curve, whereas layer-wise HP leads to more stable transfer.

**Layer-wise Learning Rate.**  One possible explanation of the "bimodal" learning rate curves in Figure 2 (duplicated in Figure 17a) is that there exist different mechanisms to lower the loss that may dominate learning at different widths and require separate HP tuning. In our two-layer ReLU network, while the first-layer neurons can rotate and grow in norm, the second-layer parameters can only affect the norm (see e.g., discussion on Wasserstein vs. Fisher-Rao dynamics for mean-field neural networks [Chi22]). There is no reason *a priori* that the optimal learning rate for the two layers coincide under mean-field / $\mu$P, hence if one layer start to contribute more to the loss decrease at certain width, we may observe a shift in the optimal learning rate that reflects the shift in the dominant mechanism (e.g., for learning non-polynomial multi-index models, training the second-layer parameters alone can achieve low loss only at *exponential* width). In such scenarios, we expect that this bimodal behavior can be remedied by the use of layer-wise learning rate.

This reasoning is empirically supported by Figure 17b where we divide the second-layer learning rate by 50. We observe that with the introduced multiplier, the learning rate curves now become roughly unimodal; moreover, the model achieves lower validation loss than the original tied learning rate setting (Figure 17a). We leave a more systematic investigation of layer-wise learning rate as future work.

### E.2 LLAMA EXPERIMENTS

For our Llama experiments we use the following Llama architecture.

| | |
|---|---|
| **$\mu$P Base Dimension** | 128 |
| **Normalization** | RMSNorm (prelayer norm) |
| **Hidden Layers** | 4 |
| **Attention Heads** | 8 |
| **Feedforward Network** | SwiGLU |
| **Feedforward Dimension** | $4\times$ Embedding Dimension |
| **Rotary Embedding** | $\theta = 10000$ |

Table 2: Llama architecture configuration used in all experiments.

We use the following schedule for the EMA to preserve performance and linearization accuracy.

**EMA Warmup** We warm up the EMA decay coefficient $\alpha_t$ linearly from $\alpha_{\text{start}} = 0.98$ to $\alpha_{\text{end}} = 0.9995$ over 2000 steps in the effective window $(1 - \alpha_t)^{-1}$. To capture early-training variation without increasing linearization error, we subsample the EMA trajectory every $\tau$ steps, with $\tau$ itself warmed up linearly from $\tau_{\text{start}} = 2$ to $\tau_{\text{end}} = 10$ over the same period.

#### E.2.1 LLAMA ADAM LR

**Llama Adam Configuration** Below is the default setup for all Llama experiments using Adam.

| | |
|---|---|
| **Dataset** | WikiText-103 |
| **Epochs** | 1 |
| **Optimizer** | Adam ($\beta_1 = 0.9,\ \beta_2 = 0.999$) |
| **Batch Size** | 128 |
| **Context Length** | 128 |
| **LR Schedule** | WSD with 4% linear warmup & 20% cooldown |

**LR Grid Search**

- LR $\in \{1.0,\ 1.4,\ 2.0,\ 2.8,\ 4.0,\ 4.8,\ 5.7,\ 6.7,\ 8.0\} \times 10^{-3}$
- $n \in \{128,\ 256,\ 512,\ 1024,\ 2048\}$

The loss $\mathcal{L}$ is evaluated on the validation split. We average runs over 3 random seeds.

#### E.2.2 LLAMA ADAM $\beta_1$ AND $\beta_2$

We repeat the same experiments from Appendix E.2.1 for the Adam $\beta$ hyperparameters, fixing LR $= 0.004$. For the $\beta_1$ experiment, we perform a sweep over $\beta_1$ with $\beta_2 = 0.999$ fixed and for the $\beta_2$ experiment we fix $\beta_1 = 0.9$ and sweep over $\beta_2$. The grids we used for the sweeps are:

- $\beta_1 \in \{0.63,\ 0.78,\ 0.86,\ 0.92,\ 0.95\}$
- $\beta_2 \in \{0.95,\ 0.98,\ 0.99,\ 0.995,\ 0.998,\ 0.9999\}$

The sweeps are performed in logspace in the effective window size $(1 - \beta)^{-1}$.

The analogs of Figures 6a, 8, 9 are presented for $\beta_1$ in Figures 18a, 19, 20 respectively and for $\beta_2$ in Figures 18b, 21, 22 respectively. We note that in this particular setting the performance is insensitive to $\beta_2$ except when it is too small. The computed values of $\hat{\kappa}(n)$ shown in Figure 23 are fairly similar for all the hyperparameters, suggesting that the dimension of this invariant subspace may be mostly

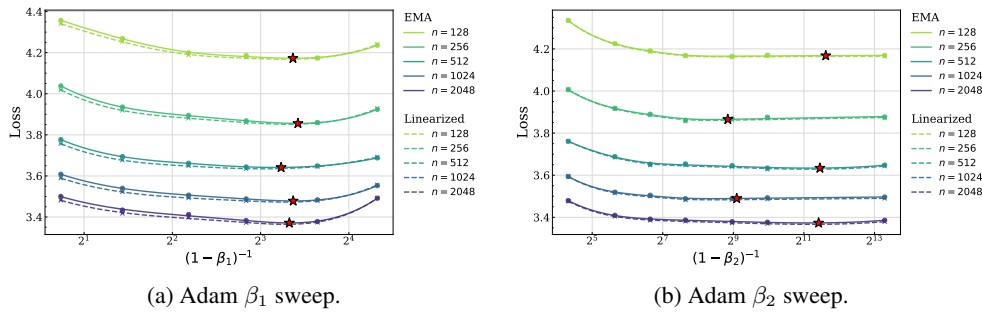

(a) Adam $\beta_1$ sweep.

(b) Adam $\beta_2$ sweep.

Figure 18: Same Llama and WikiText setup as Fig. 6, but sweeping Adam $\beta_1$ and $\beta_2$. **Left:** Smaller $\beta_1$ values increase the linearization error. **Right:** The linearization error is small for all $\beta_2$. For large enough $\beta_2$ the loss is near-optimal and transfer is nearly perfect up to harmless fluctuations in the optimal $\beta_2$.

data and architecture dependent. It will be interesting to perform similar experiments for other HPs such as weight decay and if our decomposition viewpoint can shed insight onto HPs which do not show fast transfer.

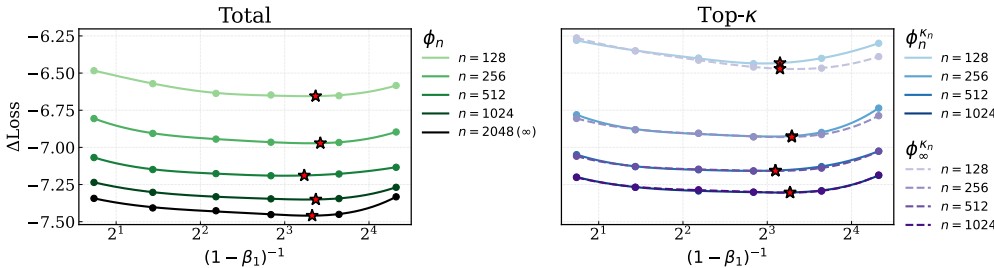

Figure 19: **Left:** Total loss curves $\phi_n$ across widths for Adam $\beta_1$. **Right:** Corresponding top-$\kappa_n$ losses $\phi_n^{\kappa_n}$ (blue dashed) and $\phi_\infty^{\kappa_n}$ (purple dashed), similar to Figure 8.

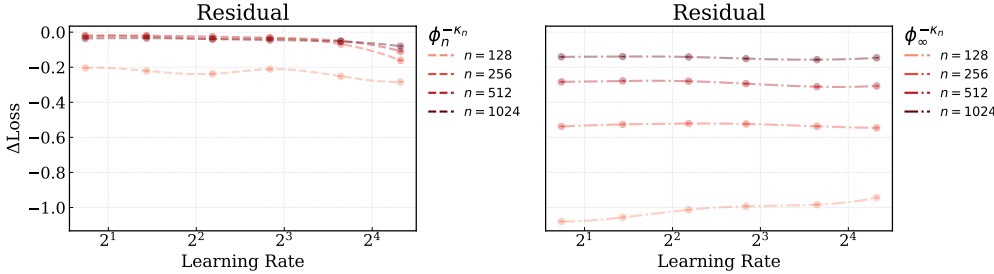

Figure 20: Residual losses around the top-$\kappa_n$ minimizers for Adam $\beta_1$, which are nearly flat as in Figure 9.

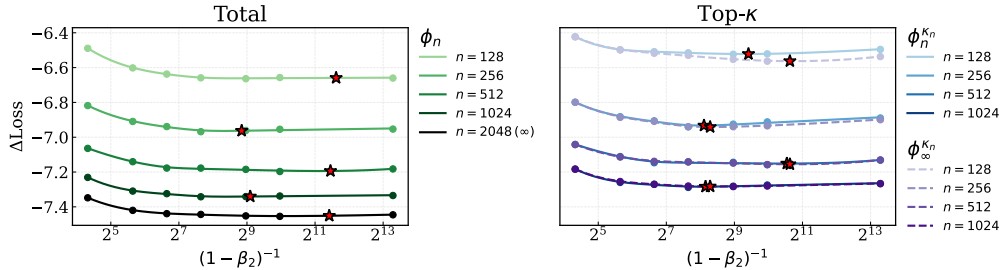

Figure 21: **Left:** Total loss curves $\phi_n$ across widths for Adam $\beta_2$. **Right:** Corresponding top-$\kappa_n$ losses $\phi_n^{\kappa_n}$ (blue dashed) and $\phi_\infty^{\kappa_n}$ (purple dashed), similar to Figure 8.

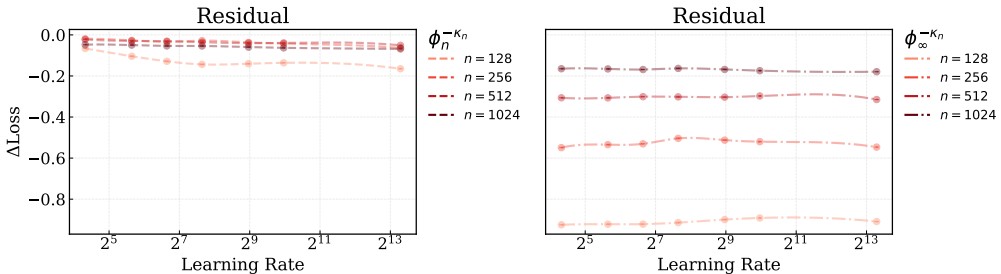

Figure 22: Residual losses around the top-$\kappa_n$ minimizers for Adam $\beta_2$, which are nearly flat as in Figure 9.

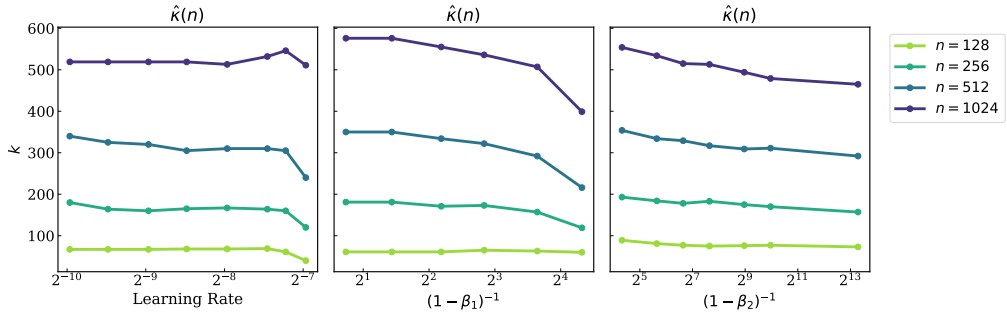

Figure 23: Computed values of $\hat{\kappa}(n)$ for sweeps over the Adam LR, $\beta_1$, and $\beta_2$.

### E.2.3 LLAMA MUON

As in Appendix E.2.1, we train a Llama-style transformer architecture with a warmup stable (WSD) learning rate schedule on WikiText-103, but using the Muon [JJB$^+$24] optimizer (see Appendix B.2) instead of Adam. The training configuration is shown below. The learning rate sweeps are shown in Figure 11a.

**Llama Muon Configuration.** Below is the setup used for the Muon training.

| | |
|---|---|
| **Dataset** | WikiText-103 |
| **Epochs** | 1 |
| **Hidden layers** | 4 |
| **Optimizer.** | Muon ($\beta = 0.95$, Adam LR= 0.004, $\beta_1 = 0.9$, $\beta_2 = 0.999$) |
| **Newton-Schulz Iterations** | 5 |
| **Batch Size** | 128 |
| **LR Schedule** | WSD with 4% linear warmup & 20% cooldown |

**LR Grid Search**

- LR $\in \{0.1, 0.2, 0.4, 0.57, 0.8, 1.1, 1.6, 1.9\} \times 10^{-2}$
- $n \in \{128, 256, 512, 1024, 2048\}$

### E.3 LLAMA DION

As in Appendix E.2.3, we train a Llama-style transformer with a warmup stable (WSD) learning rate schedule on WikiText-103, but using the Dion [AXA$^+$25] optimizer (see Appendix B.3). The training configuration is shown below. We consider two different configurations of the Dion rank hyperparameter $r$. In the **bounded rank** setting we set $r = \min(128, n/2)$ and in the **proportional rank** setting we take $r = n/2$. The learning rate sweeps for the bounded rank setting is shown in Figure 24. As we can see, the transfer is nearly perfect. On the other hand, in the proportional rank setting shown in Figure 25, we see the same imperfect transfer that we saw for Muon, but also that the performance at large width benefits from larger $r$. From Figure 26a, we see that in the bounded rank setting, the $\phi_n^k$ curves look more closer to the Adam profile in Figure 10. In particular, for $n \in \{1024, 2048\}$ the curves are nearly overlapping. However, there is still not the strong top-$k$ invariance that we saw with Adam which holds across all widths, suggesting that there may be a different notion of invariance for Dion. In contrast, in Figure 26b, we see that the $\phi_n^k$ curves strongly resemble those of Muon in Figure 12. It is an interesting future work to pin down a more appropriate notion of invariance and to understand what causes the imperfect transfer for Muon and proportional rank Dion.

| | |
|---|---|
| **Dataset** | WikiText-103 |
| **Epochs** | 1 |
| **Hidden layers** | 4 |
| **Optimizer** | Dion ($\mu = 0.95$, Adam LR= 0.004, $\beta_1 = 0.9$, $\beta_2 = 0.999$) |
| **Newton-Schulz Iterations** | 5 |
| **Batch Size** | 128 |
| **LR Schedule** | WSD with 4% linear warmup & 20% cooldown |

- LR $\in \{0.1, 0.2, 0.4, 0.57, 0.8, 1.1, 1.6, 1.9\} \times 10^{-2}$
- $n \in \{128, 256, 512, 1024, 2048\}$

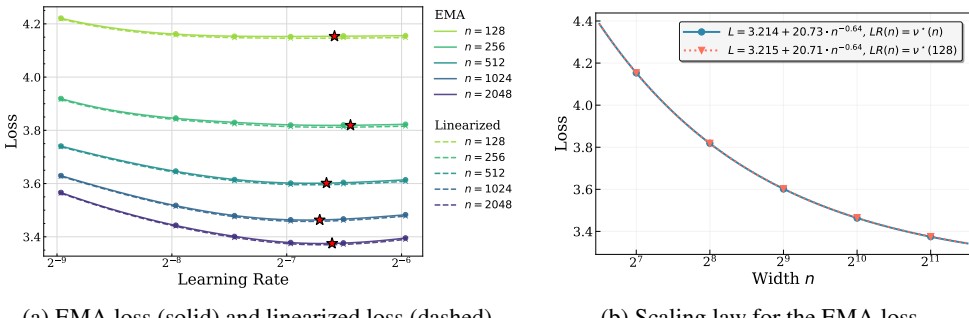

(a) EMA loss (solid) and linearized loss (dashed).

(b) Scaling law for the EMA loss.

Figure 24: Same model and dataset as Fig. 6, but trained with Dion using a bounded rank $r = \min(128, n/2)$. **Left:** EMA and linearized losses coincide. **Right:** EMA loss versus width $n$ for the learning rate choices $\nu^\star(n)$ [blue dots] and $\nu^\star(128)$ [orange triangles]. The transfer is nearly perfect in contrast with Muon (Fig. 11) and Dion with proportional rank $r = n/2$ (Fig. 25).

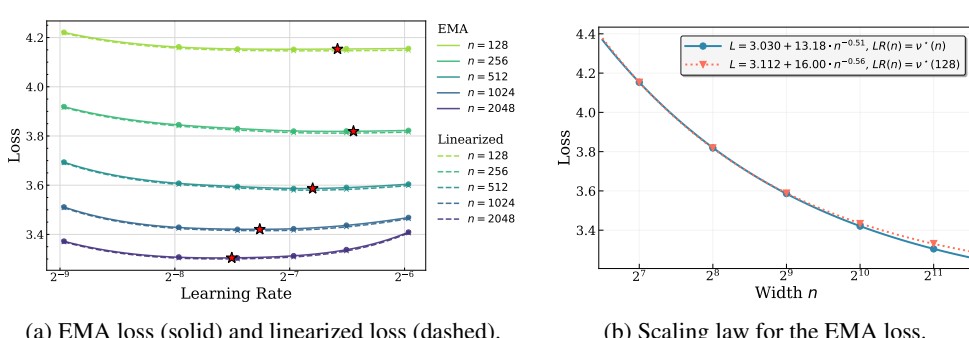

(a) EMA loss (solid) and linearized loss (dashed).

(b) Scaling law for the EMA loss.

Figure 25: Same model and dataset as Fig. 6, but trained with Dion using proportional rank $r = n/2$. **Left:** EMA and linearized losses coincide. **Right:** EMA loss versus width $n$ for the learning rate choices $\nu^\star(n)$ [blue dots] and $\nu^\star(128)$ [orange triangles]. The transfer is imperfect, similar to Muon (Fig. 11) and in contrast with Dion with bounded rank $r = \min(128, n/2)$ (Fig. 24).

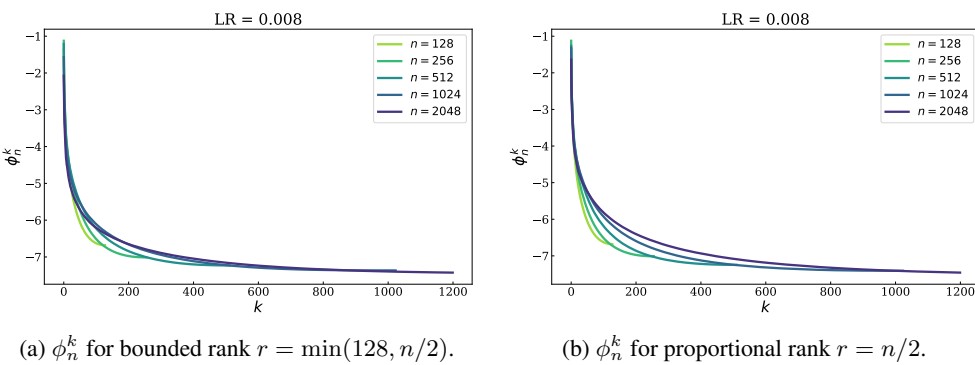

(a) $\phi_n^k$ for bounded rank $r = \min(128, n/2)$.

(b) $\phi_n^k$ for proportional rank $r = n/2$.

Figure 26: **Left:** The top-$k$ loss curves $\phi_n^k$ are more invariant in the bounded rank setting, but still appear qualitatively different from Adam and SGD. **Right:** The top-$k$ loss curves $\phi_n^k$ in the proportional rank setting look qualitatively similar to those for Muon (Fig. 12).

### E.4 GPT-2 EXPERIMENTS

For our GPT-2 experiments we use the following architecture with trainable position embeddings.

| | |
|---|---|
| **$\mu$P Base Dimension** | 128 |
| **Normalization** | RMSNorm (prelayer norm) |
| **Hidden Layers** | 4 |
| **Attention Heads** | 8 |
| **Feedforward Network** | GeLU |
| **Feedforward Dimension** | $4\times$ Embedding Dimension |

Table 3: GPT-2 architecture configuration used in all experiments.

**EMA Warmup**  We warm up the EMA decay coefficient $\alpha_t$ linearly from $\alpha_{\text{start}} = 0.995$ to $\alpha_{\text{end}} = 0.996$ over 2000 steps in the effective window $(1 - \alpha_t)^{-1}$. We subsample the EMA trajectory every $\tau$ steps, with $\tau$ itself warmed up linearly from $\tau_{\text{start}} = 1$ to $\tau_{\text{end}} = 2$ over the same period.

#### E.4.1 GPT-2 ADAM LR

Similar to the experiments in Section 4.2, we investigate transfer of the Adam LR, but using the GPT-2 architecture in Table 3 and the FineWeb dataset.

**GPT-2 Adam Configuration**  Below is the setup for GPT-2 experiments using Adam.

| | |
|---|---|
| **Dataset** | FineWeb |
| **Tokens** | 1B |
| **Optimizer** | Adam ($\beta_1 = 0.9,\ \beta_2 = 0.95$) |
| **Batch Size** | 256 |
| **Context Length** | 1024 |
| **LR Schedule** | WSD with 4% linear warmup & 20% cooldown |

**LR Grid Search**

- LR $\in \{0.3,\ 0.6,\ 0.8,\ 1.2,\ 1.7,\ 2.4,\ 3.4\} \times 10^{-2}$
- $n \in \{128,\ 256,\ 512,\ 1024,\ 2048\}$

The loss $\mathcal{L}$ is evaluated on the validation split. We average runs over 3 random seeds.

The analogs of Figures 6, 8, 9 are presented for our GPT-2 setup in Figures 27, 29, 30 respectively. As observed in the Llama experiments, we can observe the faithfulness of the linearization and nearly perfect transfer in Figure 27. In Figures 29 and 30 we see that top-$k$ invariance and residual flatness hold approximately. Although the decomposition is not as clean as it was for the Llama architecture, the trends are still suggestive and provide qualitative support for our overall conjecture. In particular, we see essentially the same qualitative profile of $\phi_n^k$ in Figure 28 which we saw for Figure 10. Interestingly however, we see that $\hat{\kappa}(n)$ is much smaller in this setting. Another difference we observe is that the top-$k$ invariance seems to break down for small learning rates and small widths in the GPT-2 setting, which did not appear to be the case for the Llama architecture. We note that the conditions for our fast transfer conjecture only require invariance to hold for nearly optimal HPs, and we speculate that invariance can indeed break for suboptimal HPs. We defer a more quantitative evaluation of this phenomenon to future work.

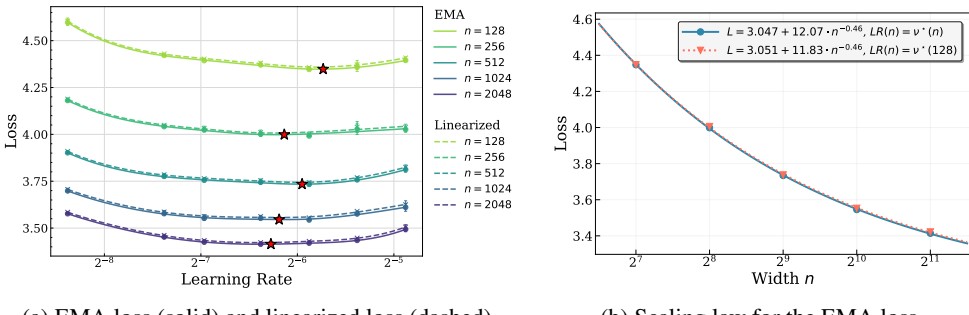

(a) EMA loss (solid) and linearized loss (dashed).  (b) Scaling law for the EMA loss.

Figure 27: Training a 4-layer GPT-2 transformer on FineWeb1B with Adam. **Left:** EMA and linearized losses nearly coincide. **Right:** EMA loss across widths under the width-dependent optimal learning rate $\nu^\star(n)$ [blue dots] and the fixed width-128 choice $\nu^\star(128)$ [orange triangles] shows near-perfect transfer across widths.

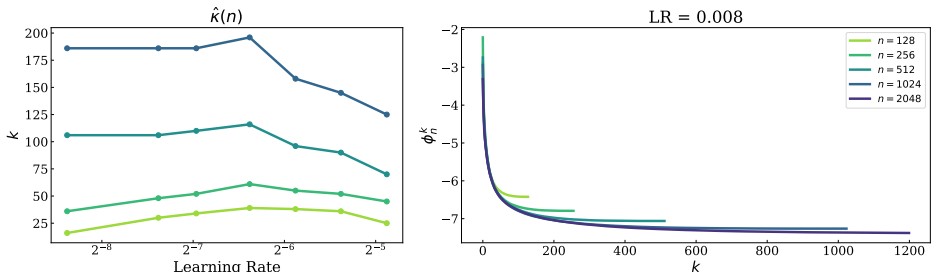

Figure 28: **Left:** Computed values of $\hat{\kappa}(n)$ using Algorithm 1. **Right:** Top-$k$ losses $\phi_n^k$ for LR = 0.008. The curves descend rapidly with $k$ and overlap over different $n$ for an intermediate range of $k$ where top-$k$ invariance holds. The profile is qualitatively similar to Figure 10.

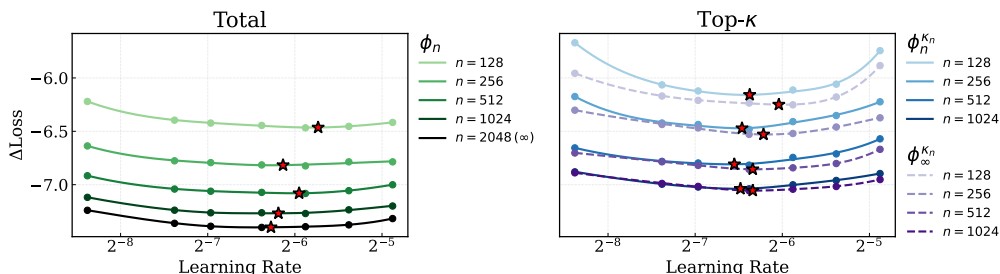

Figure 29: **Left:** Total loss curves $\phi_n$ across widths for GPT-2 experiments with Adam. **Right:** Corresponding top-$\kappa_n$ losses $\phi_n^{\kappa_n}$ (blue dashed) and $\phi_\infty^{\kappa_n}$ (purple dashed), similar to Figure 8.

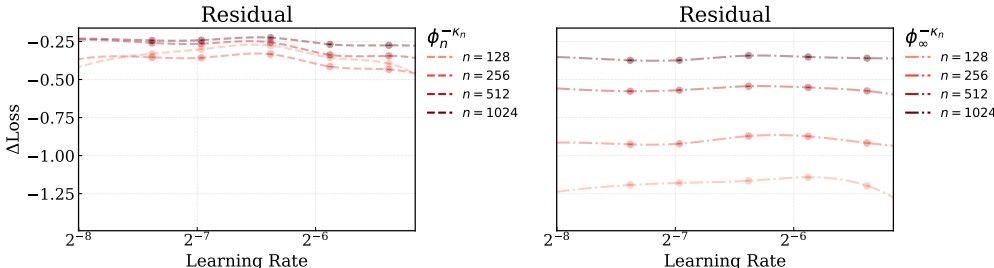

Figure 30: Residual losses around the top-$\kappa_n$ minimizers for GPT-2 with Adam, which are nearly flat as in Figure 9.

### E.4.2 GPT-2 MUON LR

Next we perform a learning rate sweep for the Muon optimizer (Appendix B.2) in our GPT-2 setup (Table 3).

**GPT-2 Muon Configuration**  Below is the setup for GPT-2 experiments using Muon.

**LR Grid Search**

| Dataset | FineWeb |
| --- | --- |
| Tokens | 1B |
| Optimizer | Muon (Adam LR = 0.01, $\beta_1 = 0.9$, $\beta_2 = 0.95$) |
| Newton-Schulz Iterations | 5 |
| Batch Size | 256 |
| Context Length | 1024 |
| LR Schedule | WSD with 4% linear warmup & 20% cooldown |

- LR $\in \{0.3, 0.6, 1.2, 2.0, 4.0, 6.0, 12.0\} \times 10^{-2}$

- $n \in \{128, 256, 512, 1024, 2048\}$

The loss $\mathcal{L}$ is evaluated on the validation split. We average runs over 3 random seeds.

The result of the sweep is shown in Figure 31 which is the analog of Figure 11. We see that although the optimal learning rate seems to shift non-trivially (Figure 31a), the Muon optimizer produces "flat" hyperparameter curves so that suboptimal learning rate does not affect the transfer performance much (Figure 31b). However, it is still possible that the transfer suboptimality will become more visible for larger widths.

In Figure 32, we plot the top-$k$ loss profiles $\phi_n^k$ for LR $\in \{0.001, 0.012\}$. We see that for the smaller learning rate the profile looks very similar to Figure 12. Interestingly however, for the larger learning rate, which is closer to the optimal learning rate, we see that the profile looks a lot closer to that observed when using the Adam optimizer. One potential explanation is that the dynamics might be more heavily dominated by the layers which use the Adam optimizer for larger learning rates. It is possible that this is linked with the stable Muon transfer observed in this setting as opposed to worse transfer performance observed in Figure 11a. More careful investigation is required to disentangle these effects, potentially by only training the layers which undergo the orthogonalized updates and ablating the number of Newton-Schulz iterations.

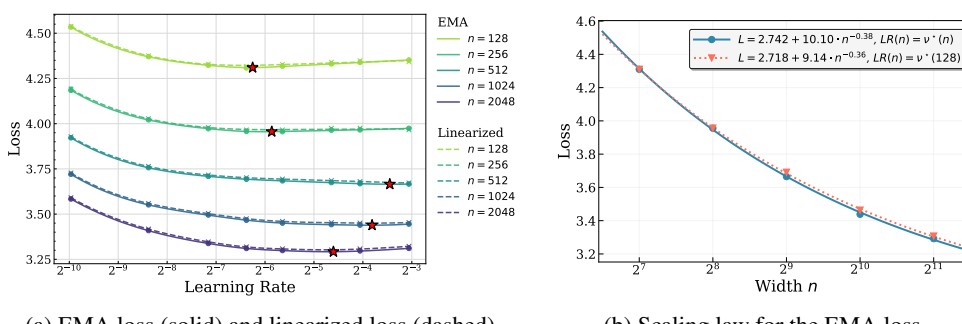

(a) EMA loss (solid) and linearized loss (dashed).      (b) Scaling law for the EMA loss.

Figure 31: Same model and dataset as Fig. 27, but trained with Muon. Although the optimal learning fluctuates across width, the insensitivity to learning rate causes transfer to be only slightly suboptimal for the given widths. **Left:** EMA and linearized losses nearly coincide. **Right:** EMA loss across widths using the two learning rate choices $\nu^\star(n)$ [blue dots] and $\nu^\star(128)$ [orange triangles].

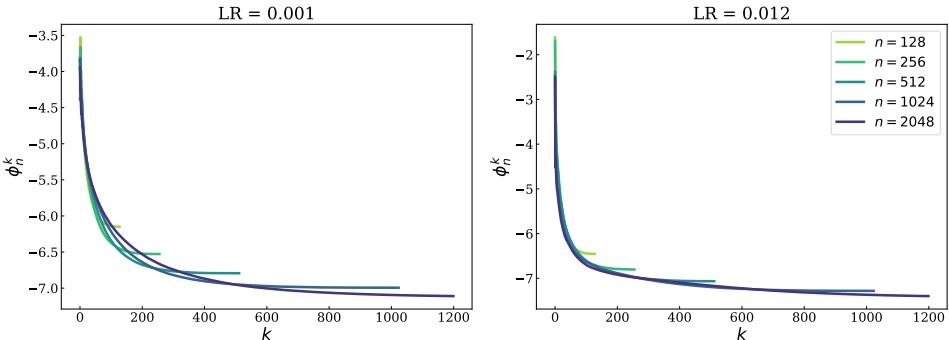

Figure 32: The qualitative behavior of the top-$k$ losses $\phi_n^k$ is learning rate dependent for GPT-2 trained with Muon. **Left:** Top-$k$ losses $\phi_n^k$ for LR $= 0.001$. The profile looks similar to the one observed for Muon on the Llama architecture in Figure 12 since top-$k$ invariance only holds for small $k$ and the descent is slow especially for large $n$. **Right:** Top-$k$ losses $\phi_n^k$ for LR $= 0.012$. The profile looks similar to those observed from the Adam optimizer (see Figures 10 and 28) since the descent is fast, there is approximate top-$k$ invariance, and the tail flattens out.

### E.5 CIFAR-10 MLP SGD

We also probe the generality of our observations to a different dataset, optimizer, and architecture: CIFAR-10 training using SGD on a 2-layer MLP with ReLU activation and no biases.

**CIFAR-10 Training Configuration** Below is the setup used for our CIFAR-10 experiment.

| | |
|---|---|
| **Dataset** | CIFAR-10 |
| **Epochs** | 100 |
| **Layers** | 2 |
| **Optimizer** | Momentum SGD ($\beta = 0.9$) |
| **Batch Size** | 512 |
| **LR Schedule** | WSD with 4% linear warmup & 20% cooldown |
| **Data Augmentation** | Mixup, Random Resized Cropping |

We use the same EMA warmup and discretization schedule as detailed in Appendix E.2. We use our largest width $n_{\max} = 8192$ as the infinite-width proxy for computing $\hat{\boldsymbol{\kappa}}(n)$ using Algorithm 1.

**LR Grid Search**

- LR $\in \{1.0, 1.4, 2.0, 2.8, 4.0, 4.8, 5.7, 6.7, 8.0\} \times 10^{-3}$

- $n \in \{128, 256, 512, 1024, 2048, 4096, 8192\}$

In the right panel of Figure 33 we see that the majority of the loss decrease occurs in the first few components, and in the left panel we see that the computed values of $\hat{\kappa}(n)$ are much smaller than the width $n$. When we apply our decomposition using the computed values of $\hat{\kappa}(n)$ in Figures 34 and 35, we see the same qualitative picture holds as in the transformer Adam experiments (for example see Figures 8 and 9 for Llama trained on WikiText-103).

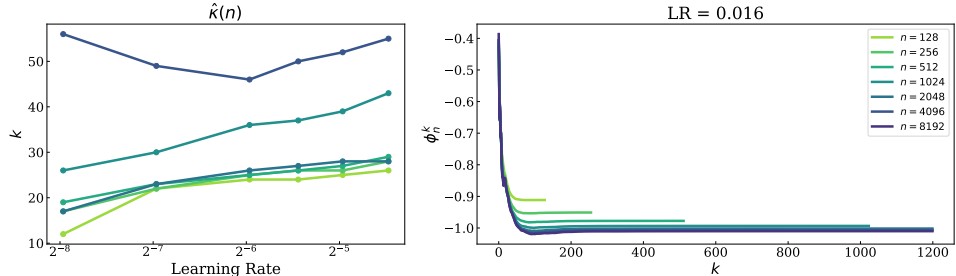

Figure 33: Two-layer MLP trained on CIFAR-10 using SGD. **Left:** Computed values of $\hat{\kappa}(n)$, with ratios $k_n/n$ much smaller than in the language setting (see Figure 10). **Right:** Top-$k$ losses $\phi_n^k$ descend with $k$ and flatten much more quickly than in Figure 10, indicating a more low-rank structure in this setting.

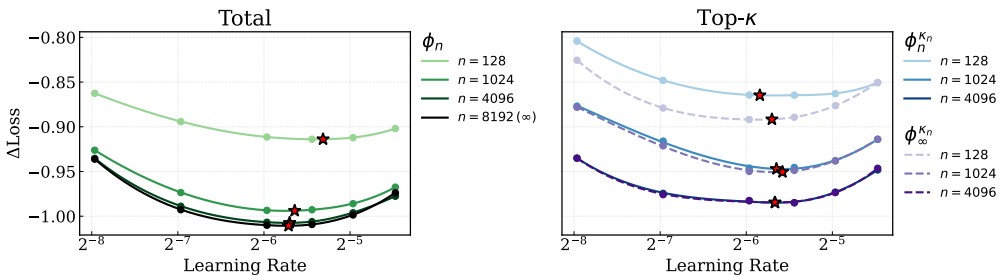

Figure 34: **Left:** Total loss curves $\phi_n$ across widths for the two-layer MLP trained with SGD. **Right:** Corresponding top-$\kappa_n$ losses $\phi_n^{\kappa_n}$ (blue dashed) and $\phi_\infty^{\kappa_n}$ (purple dashed), similar to Figure 8.

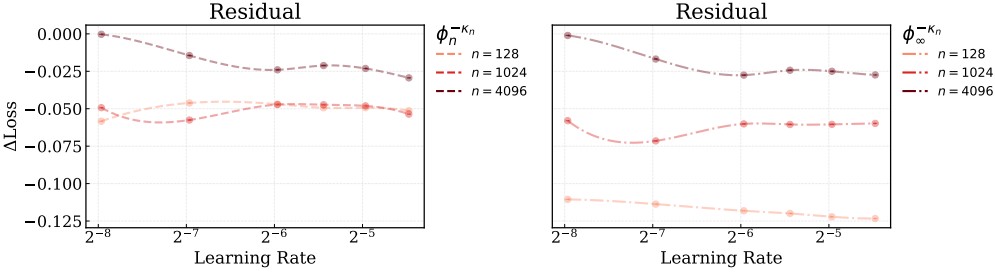

Figure 35: Residual losses around the top-$\kappa_n$ minimizers for two-layer MLP trained with SGD, which are approximately flat relative to the curvature of the top-$\kappa_n$ loss, as in Figure 9.

