# OpenReview forum: "Understanding the Mechanisms of Fast Hyperparameter Transfer"
_ICLR.cc/2026/Conference — ICLR 2026 Poster_

### Official Review · Reviewer_VXBW · 2025-10-29

**Soundness:** 3
**Presentation:** 3
**Contribution:** 4
**Rating:** 8
**Confidence:** 2

**Summary:**

The paper develops a formal framework for understanding fast hyperparameter transfer. Fast HP transfer means that hyperparameters tuned on a smaller model (less width) transfer to bigger models (more width). They give theoretical reasoning why this works, for example, in muP. They decompose the loss into top-k components and residual loss. The top-k components determine the optimal HPs and converge quickly across widths, which explains the fast HP transfer. This is validated through experiments on transformers and MLPs.

**Strengths:**

In summary:

* The paper is well-written and nicely put into the context of the related work.
* The framework of HP transfer is novel and provides interesting novel insights and theory on the topic.
* The idea of the loss decomposition seems to be new as well and quite useful, possibly also for other applications.
* The paper gives theoretical and empirical reasoning for why the HP transfer works.
* They also verify their theory on real-world models in their experiments, next to analytical ones.
* Understanding HP transfer is relevant given the trend to very large models, which can be tuned more computationally efficiently with HP transfer.

**Weaknesses:**

* It would be nice if the authors introduced the concept of muP in the introduction, since your paper is based on it. For someone not being super familiar with the topic, it needed quite a bit of effort to get into the topic.
* Multi-Fidelity optimization is also a type of HP transfer. I would appreciate a discussion on the relation to multi-fidelity as a fairly established approach.
* Please double-check for introducing abbreviations before using them.

**Questions:**

* How is k for top-k chosen in practice? What are the implications of it?
* What effect would a negative change in loss have?
* For which other HPs other than learning rate does your framework hold? Is your framework also applicable to other scale factors like depth, data, or number of training epochs?

---

> ### Author Response · Authors · 2025-11-22
>
> We thank the reviewer for their enthusiastic support and for finding our framework "novel" and capable of providing "interesting novel insights." We appreciate the acknowledgment that our paper provides both theoretical and empirical reasoning for why HP transfer works.
>
> **Intro to $\mu$P:** Thank you for the suggestion. We have added a short introduction to muP in the first paragraph of the introduction.
>
> **Multi-Fidelity Optimization:** Indeed, HP transfer can be viewed at a high level as similar to multi-fidelity optimization (MFO). The smaller proxy model provides a signal for the optimal HPs of larger models. However, HP transfer focuses on zero-shot transfer of optimal HPs across model sizes, whereas classic MFO approaches focus on finding the best hyperparameter configuration through optimal allocation of training time. The principles of MFO primarily aim to preserve the ranking of configurations and adaptively allocate resources to the most promising candidates. In this sense, we view hyperparameter transfer and MFO techniques as largely complementary.
>
> **Abbreviations**: Thank you for pointing this out, we have checked more carefully for this issue.
>
> **Choosing $k$**: The process for choosing $k$ is detailed in Appendix C. At a high-level, since we can record $\phi_k$ for any $k$, we explicitly search for $k$ post-hoc such that we can see top-$k$ invariance and residual flatness by optimizing for these properties. Our aim is to show that such a $k$ exists, thus demonstrating the conjectured structure under which fast HP transfer is possible. To our knowledge, this provides the first empirical evidence that the optimal HP is approximately determined by a small set of statistics that converges faster than statistics that decide the loss scaling.
>
> **Negative Loss Change**: Perhaps we have misunderstood your question, but a negative change in loss $\Delta \mathcal{L} = \mathcal{L}_T - \mathcal{L}_0$ just means that the final loss is smaller than the initial loss.
>
> **Other HPs**: Please see our General Response.
>
> We would be happy to answer any follow-up questions in the discussion period.

---

> > ### Comment · Reviewer_VXBW · 2025-11-25
> >
> > Thank you for your answers. I was already very positive about your paper and will stay with that opinion.

---

> > > ### Author Response · Authors · 2025-11-25
> > >
> > > Thank you again for your positive assessment of our work, please let us know if we can provide any further clarifications that would increase your confidence score

---

### Official Review · Reviewer_PzsK · 2025-10-31

**Soundness:** 3
**Presentation:** 3
**Contribution:** 2
**Rating:** 6
**Confidence:** 2

**Summary:**

The work introduces a conceptual framework to study the transfer properties of hyperparameters of neural networks from low to large scales. A decomposition of the loss suggests a connection of certain optimization statistics to the convergence properties of optimal hyperparameter settings.

**Strengths:**

* Understanding the mechanisms behind hyperparameter transfers from low to large scales is an important issue to study as it can inform future algorithmic directions
* Assumptions are clearly stated
* Existing literature on analysis of hyperparameter transfer across scales and related analysis methods is well covered

**Weaknesses:**

* Unclear if there are any direct practical implications of the introduced conceptual framework
* In line with previous work, the conceptual framework targets a very specific choice of hyperparameters and optimization settings.
* Code to reproduce the experiments is not provided, although some details on the trainings and fixed hyperaparamter settings are provided. What are the specifics of the "Llama-style" transformer architecture you used?

**Questions:**

* The authors focus on a very specific choice of hyperparameters. They state "however our framework can be used more broadly for reasoning about scale-aware HPs", could you elaborate on that? Where are the limitations of your framework?

* What scale in terms of # of trainable parameters do your width choices correspond to?

Minor comments
* Numbering all equations would be good to facilitate discussions
* Axis in Figures descriptions are very small

---

> ### Author Response · Authors · 2025-11-22
>
> We thank the reviewer for their constructive feedback and for highlighting the importance of understanding the mechanisms behind HP transfer. We are glad the reviewer found our assumptions clearly stated and the literature well covered.
>
> **Practical Implications:** Please see our General Response.
>
> **Specific HPs and Limitations:** Please see our General Response.
>
> **Reproducibility:** Thank you for the suggestion. We have added full architectural details for the "Llama-style" transformer in Appendix D.2.
>
> **Trainable parameters:** Below is the table mapping the widths used in our experiments to trainable parameter counts
>
> | Wdith | Parameters |
> |------------|------------------------|
> | 128        | 13.93M   |
> | 256        | 29.95M   |
> | 512        | 68.29M   |
> | 1024       | 170.13M |
> | 2048       | 474.48M |
>
> We would be happy to answer any follow-up questions in the discussion period.

---

### Official Review · Reviewer_WG5j · 2025-10-31

**Soundness:** 2
**Presentation:** 2
**Contribution:** 3
**Rating:** 4
**Confidence:** 2

**Summary:**

The paper explores the gap in understanding of what determines the fast convergence rate of zero-shot hyperparameter transfer at the infinite-width limit using $\mu$Parameterization, as seen empirically.
The authors present a novel decomposition of the loss trajectory through a linearization scheme over an EMA of the original trajectory.
They show that the sum of the top-k eigenvalues of the gradient matrix captures the loss change in the top-k directions of maximum change in loss.
This measure can be seen to be invariant across width-scaling, whereas the other residual components appear to reflect scaling with width.
The authors posit that this indicates the existence of a subspace of the trajectory (as captured by EMA smoothing), such that tuning for those top-k components at a width $n$ can guarantee fast hyperparameter transfer as $n \rightarrow
\infty$.

**Strengths:**

* Novel framework and formulation to try and explain the practical gains of an important theoretical finding in $\mu$Parameterization.

**Weaknesses:**

* Hard to follow the notations in Section 4, especially with the incremental evolution of $\phi$ and its usage.

* Hard to see the practical implications of the finding, especially with the requirement of adjusting $k$ for any new task. Unless the only goal was to *explain* why fast HP transfer shows up in practice, the claim in abstract for validating this for LLMs might need to be revised.

* The understanding of the *main* paper on its own is a bit hard, with adequate parsing of the Appendix details quite crucial to sometimes connect points made in the main paper.

**Questions:**

Below are the enumerated questions and suggestions.

Please note that the rating will be increased contingent on the points below being addressed/clarified.

1\. The authors could consider explaining in brief the comparative scaling of the hyperparameter value compared to the performance metric with width, as the meaning of "fast transfer", in the abstract.

2\. L60-63: The authors could consider referring to the relevant sections.

3\. L116: In comparison to previous literature, is one of the contribution of this paper, to show that the top-k eigenvalues of the gradient matrix can adequately capture invariance across appropriate width-scaling?

4\. L172:173: Does this specifically apply to only certain types of hyperparameters? Or only the abcd-parameterization parameters that are scaled in width?

5\. L266-269: Could the authors explain why this happens and its implications?

6\. L255-266: *Top-$\kappa$ strong convexity*, here both variables appear same, $\phi_{n}^{\kappa_n}$ . If $\kappa(\nu)$ is an integer related to width-$n$, then what does the super- and sub-script denote?

7\. Figure 4.b): Can this be explained more please? Especially the role or choice of `Step 7185`. What is the blue dot line, in $LR(n) = \nu^{*}(n)$?

8\. Figure 7 (right): Could different colours be used here?

9\. L460-464: Is flatness of the loss residual the issue here or the loss gap at $n \rightarrow \infty$?

---

> ### Author Response · Authors · 2025-11-22
>
> We thank the reviewer for their detailed feedback and for recognizing our framework and formulation as "novel" and useful for explaining practical gains in $\mu$Parameterization.
>
> **Notation in Section 4:** Following your suggestions, we have significantly revised Section 4.1 to aid clarity. We added a notation table (Table 1) to explicitly map the usage of $\phi$ to their equations.
>
> **Practical Implications:** Please see our General Response, where we detail our contributions as the first formal framework for HP transfer and outline specific practical utilities
>
> **Parsing Main Paper**: We hope the revisions in Section 4, along with the new notation table and clarified definitions, have improved the readability of the main paper.
>
> **Fast Transfer Definition in Abstract:** This is a good suggestion, we have added this into the abstract.
>
> **L60-63:** We have added specific references to Section 4.1 and Figure 1.
>
> **Previous Literature:** Yes, a primary contribution of this paper is to precisely state this claim, and then to measure and validate it empirically. The invariance idea is indirectly suggested by theoretical work showing a spike + bulk decompositions of the gradient for various simple tasks. Note that compared to prior works (e.g., [Noci et al. 2024]) empirically showing that top eigenvalues of the Hessian converges rapidly with scale, our decomposition has a much more direct tie to HP transfer as we explicitly track the loss decrease across components (on the time-averaged trajectory). We have made this connection more explicit in the revision.
>
> **Other HPs**: All the definitions and claims made in Section 3.1 are part of a general formalism which applies to arbitrary hyperparameters and scaling dimensions. For example, it covers the ridge HP in random features regression discussed in Section 3.2. Please see the general response.
>
> **L266-269:** This is a good question and it is difficult to exactly understand what is happening (e.g. derive the rates theoretically) . At a high-level, we know from prior theoretical results [Safran & Lee 2022] that the number of neurons must be exponential in the dimension in order to solve this task, so it is plausible that additional neurons affect the dynamics significantly and break the low-rank structure we posit is useful for fast HP transfer. A speculative implication is that more generally tasks where approximation is difficult and is the primary bottleneck at polynomial scale will suffer from slower HP transfer.
>
> **L255-266**:  We apologize for the confusion. We have clarified this in the revision (i.e., the new Table 1). $\phi_n$ denotes the loss of a width-$n$ model. $\kappa(\nu)$ is the truncation index function. Therefore, $\phi_n^{\kappa_n}(\nu)$ refers to the top-$k$ loss component of a width-$n$ model, where the index $k$ is chosen by $\kappa_n(\nu)$
>
> **Figure 4b**: "Step 7185" refers to the training step (one epoch) at which losses were recorded. The blue dotted line represents the loss using the optimal learning rate tuned specifically for each width. The orange dotted line represents the loss using the optimal learning rate from width 128 transferred to larger widths. We have updated the caption to clarify this.
>
> **Figure 7 Colors**: We have updated the colors to aid visualization.
>
> **L460-464:** We are not quite sure what the question is asking, would you mind elaborating?
>
> We would be happy to answer any follow-up questions in the discussion period.

---

> > ### Comment · Reviewer_WG5j · 2025-11-27
> > **Final Reviewer Comment**
> >
> > I thank the authors for answering the questions posed and incorporating the relevant suggestions, too.
> >
> > My immediate queries have been answered, and the empirical evaluations (Appendix) have been made stronger.
> >
> > The score has been increased.
> >
> > Thank you for the discussion and all the best!

---

### Official Review · Reviewer_ssQX · 2025-11-03

**Soundness:** 3
**Presentation:** 4
**Contribution:** 3
**Rating:** 8
**Confidence:** 1

**Summary:**

Scaling-aware hyperparameters propose hyperparameter tuning with a small-scale model (e.g., small width/depth) and then directly use the tuned hyperparameter (after modification with certain rules) for training a large-scale model. Recently, Maximal Update Parameter ($\mu P$) proposed a theoretical framework for scaling-aware hyperparameter tuning when scaling model width, but in the mean-field regime only. However, such fast-hyperparameter transfer phenomenon also works with finite width/data as well, but there is no prior explanation/justification for this. This is the main motivation of this work.

To explain fast hyperparameter transfer in width-scaling, the authors propose a new decomposition of the optimization trajectory into two components: (1) a component that converges quickly with model width and determines the optimal hyperparameters, and (2) a component that improves the loss when the width is increased but has a negligible impact on HP choice. The authors then hypothesize that this one might explain the hyperparameter transfer with width scaling, and then validate their finding with empirical experiments.

**Strengths:**

The paper is very well-written and easy to follow. I hardly see any typos. The motivation is well-presented and very clear. Though I am not really an expert in this field, I can follow the paper very easily. I think it is a good paper.

**Weaknesses:**

First things first, since I am not an expert in this field, my opinion might be biased and not fairly evaluate the contribution of this paper.

1. One might argue that the paper only focuses on fast hyperparameter transfer in terms of width. In practice, one might want to know about fast hyperparameter transfer when scaling the depth instead. However, I would not consider it an issue, since it is the weakness of the Tensor Program series in general. Besides, I think that the setting is good enough, since we can simply fix the depth, train with minimum width, and it should be fast enough.

Since I am not an expert, I can only give support for this paper with a low confidence score. I will leave the decision to the AC and other expert reviewers.

**Questions:**

See weaknesses

---

> ### Author Response · Authors · 2025-11-22
>
> We thank the reviewer for their strong support and for finding the paper "very well-written" and the motivation "clear." We appreciate the recognition of the paper's ease of following despite the technical content.
>
> **Response to Weaknesses**: We agree with the reviewer that the question of fast hyperparameter transfer beyond width is fascinating and believe that it is a great direction for future work. As noted, it is already practically useful to understand the width scaling setting, since HP transfer across width can drastically save compute for HP tuning. We view our work as establishing the necessary tools to tackle these broader settings in the future.
>
> We would be happy to answer any follow-up questions in the discussion period.

---

> > ### Comment · Reviewer_ssQX · 2025-11-25
> > **Official Comment by Reviewer ssQX**
> >
> > I thank the authors for the answer. My evaluation remains the same.

---

### Author Response · Authors · 2025-11-22
**General Response**

We thank all reviewers for their thoughtful and constructive feedback.
Below we summarize and respond to two broad themes that appeared across multiple reviews.
Please see our individual responses for replies to reviewer-specific questions and suggestions.

**Scope of HPs and scaling dimensions**: Our framework and the theoretical results presented in Section 3.1 are general and agnostic to the choice of hyperparameters and scaling dimension. The top-$k$ decomposition introduced in Section 4.1 is a specific tool for understanding transfer when the scaling dimension is the model width, but it can be used to study any hyperparameter. We choose to focus on width scaling since this is the original and most well-established scaling dimension for hyperparameter transfer. We choose to focus mainly on the learning rate since this is empirically the most important HP to tune in practice and has been the main application of $\mu$-transfer. However, in Appendix D.4 we also look at the Adam $\beta_1$ and $\beta_2$ HPs. For future work, it will be interesting to validate the conclusions on other HPs of interest such as weight decay and to study other scaling dimensions such as depth. We view our contribution as laying the groundwork for such follow-up investigations.

**Contributions and Practical Implications**: The contribution of our work is two-fold: (i) to formalize the notion of fast and useful HP transfer, and (ii) to posit and empirically validate a plausible mechanism via trajectory decomposition. For (i), we introduce, to our knowledge, the first mathematical framework to reason about HP transfer (Propositions 1 and 2), as well as the first provable example where HP transfer yields computational gain (Proposition 3). For (ii), we present extensive empirical evidence that our proposed top-$k$ loss decomposition explains fast transfer. While the optimal $k$ is task-dependent, our experiments demonstrate that there exists $k \ll n$ where the top-$k$ trajectory approximately decides the optimal HP, supporting our conjecture. Building on these foundations, we believe our study suggests several practical directions:

1. **Diagnostic Tool**: Measuring the top-$k$ loss provides a diagnostic for determining if transfer is occurring properly. One could examine plots like those in Figure 6 to check if the transfer is reasonable. Future work can convert this into a quantitative test.

2. **Improving Transfer**: Our insights might be used to improve transfer algorithms by explicitly enforcing top-$k$ invariance during the tuning process.

3. **Predicting Failure Modes**: Our results provide a conceptual model for predicting when transfer will fail. For example, given that OOD loss depends strongly on the tail components of our decomposition due to the training-test misalignment, our framework would predict that using OOD loss as the transfer metric may lead to unreliable HP transfer.

---

### Meta-Review · Area_Chair_jzCT · 2026-01-06

**Summary:**

This paper presents a formal framework for understanding fast hyperparameter transfer across model width, and proposes a novel loss trajectory decomposition to explain why optimal hyperparameters often stabilize faster than performance metrics under scaling. The work combines theoretical analysis with extensive empirical validation on synthetic settings, MLPs, and LLM-style transformers.

The submission received mostly positive reviews. Reviewers highlighted the novelty of the framework, the clarity of motivation, and the strength of the empirical evidence. One reviewer initially raised concerns about notation clarity and practical implications, but these were adequately addressed in the rebuttal and revisions, leading to an increased score. Remaining concerns mainly relate to scope (focus on width scaling and learning rate), which are reasonable limitations rather than fundamental weaknesses.

Overall, the paper offers a solid conceptual contribution to understanding scale-aware hyperparameters and provides a plausible mechanism for fast transfer observed in practice. I recommend acceptance.

**Reviewer Concerns:**

Addressed concerns: notation clarity and readability; architectural and experimental details.

Outstanding or partially outstanding concerns: scope of scaling dimensions and hyperparameters; direct algorithmic impact.

**Reviewer Scores:**

Reviewer WG5j: Score increased from 4 to 6, as explicitly stated by the reviewer after the rebuttal.

---

### Decision · Program_Chairs · 2026-01-26

Accept (Poster)